# Murine muscle stem cell response to perturbations of the neuromuscular junction are attenuated with aging

Jacqueline A Larouche[1,2†], Mahir Mohiuddin[3,4,5†], Jeongmoon J Choi[3,4,5†], Peter J Ulintz[1,2,6], Paula Fraczek[1,2], Kaitlyn Sabin[1,2], Sethuramasundaram Pitchiaya[7], Sarah J Kurpiers[1,2], Jesus Castor-Macias[1,2], Wenxuan Liu[8,9,10], Robert Louis Hastings[11,12], Lemuel A Brown[13], James F Markworth[13], Kanishka De Silva[1,2], Benjamin Levi[14,15,16], Sofia D Merajver[1,6], Gregorio Valdez[11,12], Joe V Chakkalakal[8,9,10], Young C Jang[3,4,5*], Susan V Brooks[1,13*], Carlos A Aguilar[1,2,15,16*]

[1]Department of Biomedical Engineering, University of Michigan, Ann Arbor, United States; [2]Biointerfaces Institute, University of Michigan, Ann Arbor, United States; [3]Parker H. Petit Institute of Bioengineering and Bioscience, Georgia Institute of Technology, Atlanta, United States; [4]School of Biological Sciences, Georgia Institute of Technology, Atlanta, United States; [5]Wallace Coulter Departmentof Biomedical Engineering, Georgia Institute of Technology, Atlanta, United States; [6]Internal Medicine-Hematology/Oncology, University of Michigan, Ann Arbor, United States; [7]Michigan Center for Translational Pathology, University of Michigan, Ann Arbor, United States; [8]Department of Pharmacology and Physiology, University of Rochester Medical Center, Rochester, United States; [9]Department of Biomedical Engineering, University of Rochester Medical Center, Rochester, United States; [10]Wilmot Cancer Institute, Stem Cell and Regenerative Medicine Institute, and The Rochester Aging Research Center, University of Rochester Medical Center, Rochester, United States; [11]Departmentof Molecular Biology, Cell Biology and Biochemistry, Brown University, Providence, United States; [12]Center for Translational Neuroscience, Robert J. and Nancy D. Carney Institute for Brain Science and Brown Institute for Translational Science, Brown University, Providence, United States; [13]Department of Molecular & Integrative Physiology, University of Michigan, Ann Arbor, United States; [14]Department of Surgery, University of Texas Southwestern, Dallas, United States; [15]Childrens Research Institute and Center for Mineral Metabolism, Dallas, United States; [16]Program in Cellular and Molecular Biology, University of Michigan, Ann Arbor, United States

*For correspondence:
young.jang@gatech.edu (YCJ);
svbrooks@umich.edu (SVB);
caguilar@umich.edu (CAA)

†These authors contributed equally to this work

**Competing interests:** The authors declare that no competing interests exist.

**Abstract** During aging and neuromuscular diseases, there is a progressive loss of skeletal muscle volume and function impacting mobility and quality of life. Muscle loss is often associated with denervation and a loss of resident muscle stem cells (satellite cells or MuSCs); however, the relationship between MuSCs and innervation has not been established. Herein, we administered severe neuromuscular trauma to a transgenic murine model that permits MuSC lineage tracing. We show that a subset of MuSCs specifically engraft in a position proximal to the neuromuscular junction (NMJ), the synapse between myofibers and motor neurons, in healthy young adult muscles. In aging and in a mouse model of neuromuscular degeneration (Cu/Zn superoxide dismutase knockout – $Sod1^{-/-}$), this localized engraftment behavior was reduced. Genetic rescue of motor neurons in $Sod1^{-/-}$ mice reestablished integrity of the NMJ in a manner akin to young muscle

and partially restored MuSC ability to engraft into positions proximal to the NMJ. Using single cell RNA-sequencing of MuSCs isolated from aged muscle, we demonstrate that a subset of MuSCs are molecularly distinguishable from MuSCs responding to myofiber injury and share similarity to synaptic myonuclei. Collectively, these data reveal unique features of MuSCs that respond to synaptic perturbations caused by aging and other stressors.

## Introduction

Skeletal muscle atrophy and weakness are primary features of physical frailty and are common among the elderly and patients afflicted with neuromuscular disorders (*Marcell, 2003*). The decline in the health and repair of skeletal muscle can be partially attributed to two key features. First, aging invokes decreases in number (*Blau et al., 2015*) and function (*Schultz and Lipton, 1982*) of a population of resident stem cells called satellite cells (*Wang and Rudnicki, 2012*; *Shcherbina et al., 2020*) or muscle stem cells (MuSCs) (*Porpiglia et al., 2017*), which are the principal source of myonuclei for muscle growth and regeneration (*Yin et al., 2013*; *Almada and Wagers, 2016*), contributing to muscle degeneration and sarcopenia. Second, impairments in innervation and synaptic integrity (*Ham et al., 2020*) between motor neurons and myofibers (*Shi et al., 2012*) become prevalent with aging (*Jang and Van Remmen, 2011*). To date, the connection between MuSCs and neuromuscular junction (NMJ) dysfunction in aging remains underexplored.

MuSCs have been shown to support the integrity of the NMJ by generating new synaptic myonuclei (SyM) (*Liu et al., 2017*), but how MuSCs sense and respond to innervation and how this behavior in old age remains poorly understood. Recent studies have shown that specific genetic depletion of MuSCs can accelerate aging-related NMJ deterioration through loss of SyM (*Liu et al., 2017*) and reciprocally, MuSC depletion impairs the regeneration of NMJs after nerve injury (*Liu et al., 2015*). Accordingly, understanding how changes in innervation influences MuSC actions such as differentiation to become SyM can provide insight into NMJ deterioration and muscle wasting observed with aging and neurodegenerative disease.

Herein, we assessed the response of MuSCs to functional and structural changes at NMJs. We demonstrate that disruption of innervation results in fusion of a subset of MuSC-derived progenitors adjacent to NMJs. We next show that after NMJ perturbation in aged muscle, MuSCs migrate toward the NMJ but do not fuse as efficiently as in young adult mice. Single cell gene expression analysis revealed a subset of MuSCs express transcripts associated with NMJs. We next show that MuSCs expressing transcripts associated with NMJs are distinct from those that respond to muscle injury and are transcriptionally similar to SyM. We next demonstrate that this subset of MuSCs is also present in an orthogonal model of neurodegeneration ($Sod1^{-/-}$ mice). Genetic rescue of motor neurons by constitutive expression of SOD1 driven by the synapsin promotor in $Sod1^{-/-}$ mice (Syn-TgSod1$^{-/-}$) attenuated NMJ degeneration, decreased the presence of MuSC subsets expressing NMJ markers and partially restored the ability of MuSCs to contribute SyM after neuromuscular trauma. Collectively, these data reveal that MuSCs respond to NMJ perturbations and motor neuron injury.

## Results

### Denervation engenders MuSC actions proximal to NMJs

We reasoned that if MuSCs exhibit a relationship with the NMJ, acute perturbations to the NMJ would engender MuSC actions proximal to this niche. We utilized a MuSC lineage tracing system (*Pax7$^{CreER/+}$-Rosa26$^{nTnG/+}$*: P7$^{nTnG}$), whereby all nuclei contain a red fluorescent protein and after administration of tamoxifen, Pax7$^+$ MuSCs and their progeny are indelibly labeled with a nuclear green fluorescent protein (nGFP, *Figure 1a*, *Liu et al., 2017*). We contrasted acute and specific perturbation of motor neurons using sciatic nerve transection (SNT), which results in complete denervation and re-innervation 4–6 weeks after injury (*Liu et al., 2017*), with intramuscular injection of barium chloride (BaCl$_2$) in extensor digitorum longus (EDL), a muscle primarily composed of type II fibers in mice. Consistent with previous observations (*Liu et al., 2017*), imaging myofibers 5 days after tamoxifen administration, only Pax7$^+$ MuSCs are initially nGFP labeled along the length of isolated myofibers, and devoid of precocious nGFP expression in myonuclei (*Figure 1b*). Examination

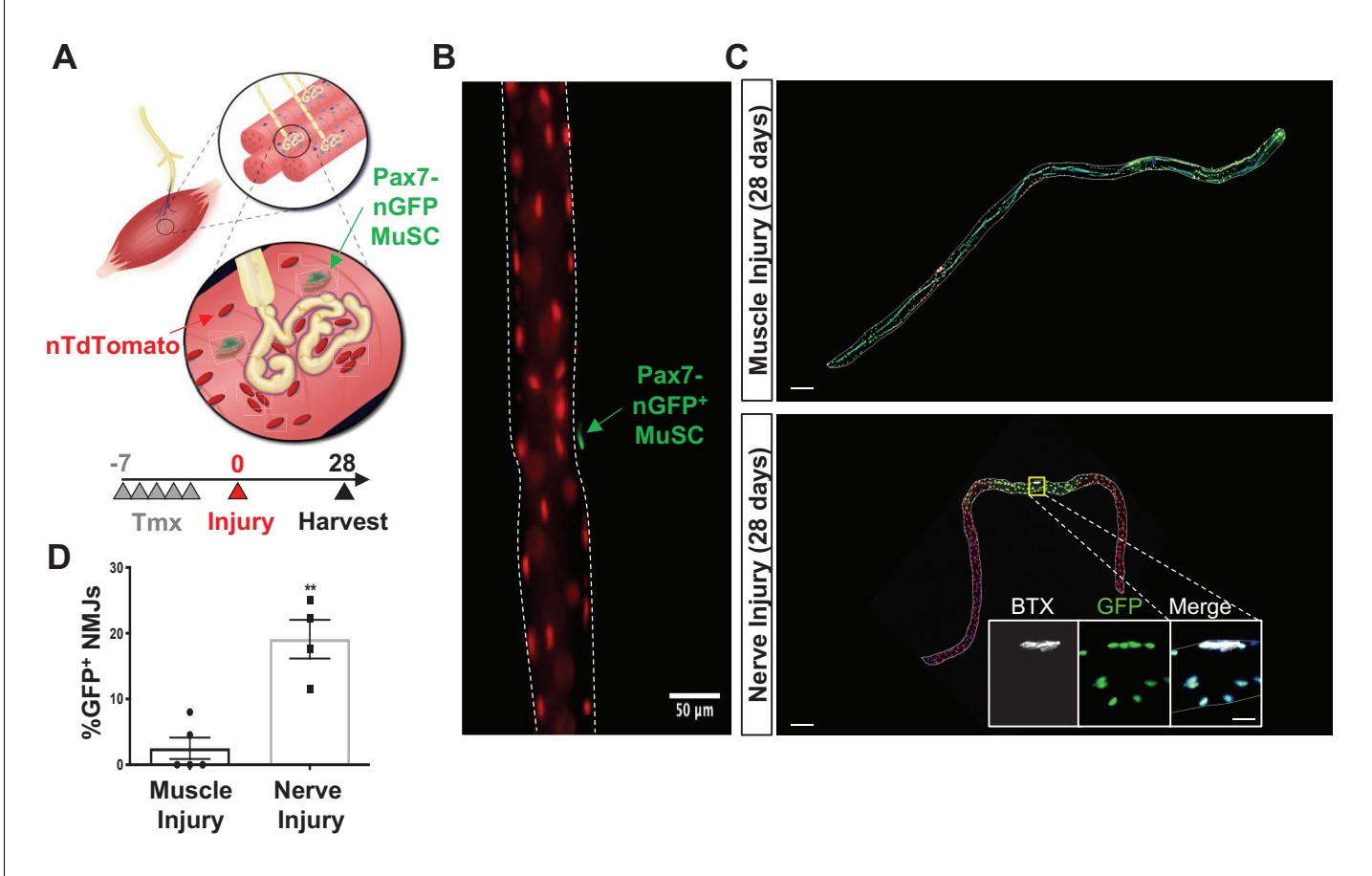

**Figure 1.** Denervation induces muscle stem cell (MuSC) actions proximal to the neuromuscular junction (NMJ). (**A**) Schematic of NMJ from Pax7$^{CreER/+}$-Rosa26$^{nTnG}$ mice, which display red fluorescent protein (RFP) in their nuclei and following administration of tamoxifen, Pax7$^+$ MuSCs (labeled with green arrow) and their progeny are labeled with a nuclear green fluorescent protein (nGFP). (**B**) Representative immunofluorescence image of single myofiber isolated from extensor digitorum longus (EDL) muscle showing GFP expression exclusively in MuSCs and red nuclei. Scale=50 μm. (**C**) After muscle injury through barium chloride (BaCl$_2$) injection (top), myofiber degeneration resulted in contribution of MuSC-derived progenitors and centrally located myonuclei along the entire length of the regenerated myofiber. After sciatic nerve transection (SNT), MuSC-derived myonuclei were confined at or near NMJ myofiber regions. Magnified inset images show NMJs (acetylcholine receptor [AChR] labeled with α-bungarotoxin [BTX]) with nGFP$^+$ nuclei. Scale bar for myofibers = 200 μm for inset = 25 μm. n = 5 muscles for both injury types and 20–30 myofibers counted from each isolated muscle. (**D**) Quantification of fraction of nGFP$^+$ nuclei underneath and near synapses compared to all positions on myofibers, where **p < 0.01 using two-sided t-test.

of single myofibers 28 days after SNT or BaCl$_2$ injection revealed variations in MuSC-derived contribution of nGFP. As expected, BaCl$_2$-induced myofiber degeneration results in contribution of MuSC-derived progenitors and centrally located myonuclei along the entire length of the regenerated myofiber (*Figure 1c*). In contrast, 28 days after SNT during re-innervation, MuSC-derived myonuclei (indicated by nGFP$^+$) were confined at or near young NMJ myofiber regions (<250 μm from the NMJ, *Figure 1c–d*). While we cannot discount that existing SyM incorporate nGFP after MuSC engraftment, diffusion of molecules near the NMJ is limited to prevent synaptic transcription in myonuclei outside of the junctional area (*Schaeffer et al., 2001*). Taken together, these results show that MuSCs are sensitive to NMJ disruptions, and in such contexts, fuse near NMJs.

## A subset of MuSCs are proximal to NMJs in young and aged muscle

Given the contribution of MuSC-derived SyM following SNT, we hypothesized that a fraction of MuSCs would be proximal to NMJs prior to injury. We utilized a MuSC fluorescent reporter that permits lineage tracing of the entire cell rather than just the nucleus (Pax7$^{CreER/+}$-Rosa26$^{TdTomato/+}$) to quantify proximity between the synapse and MuSCs (*Figure 2a–b*). After administration of tamoxifen

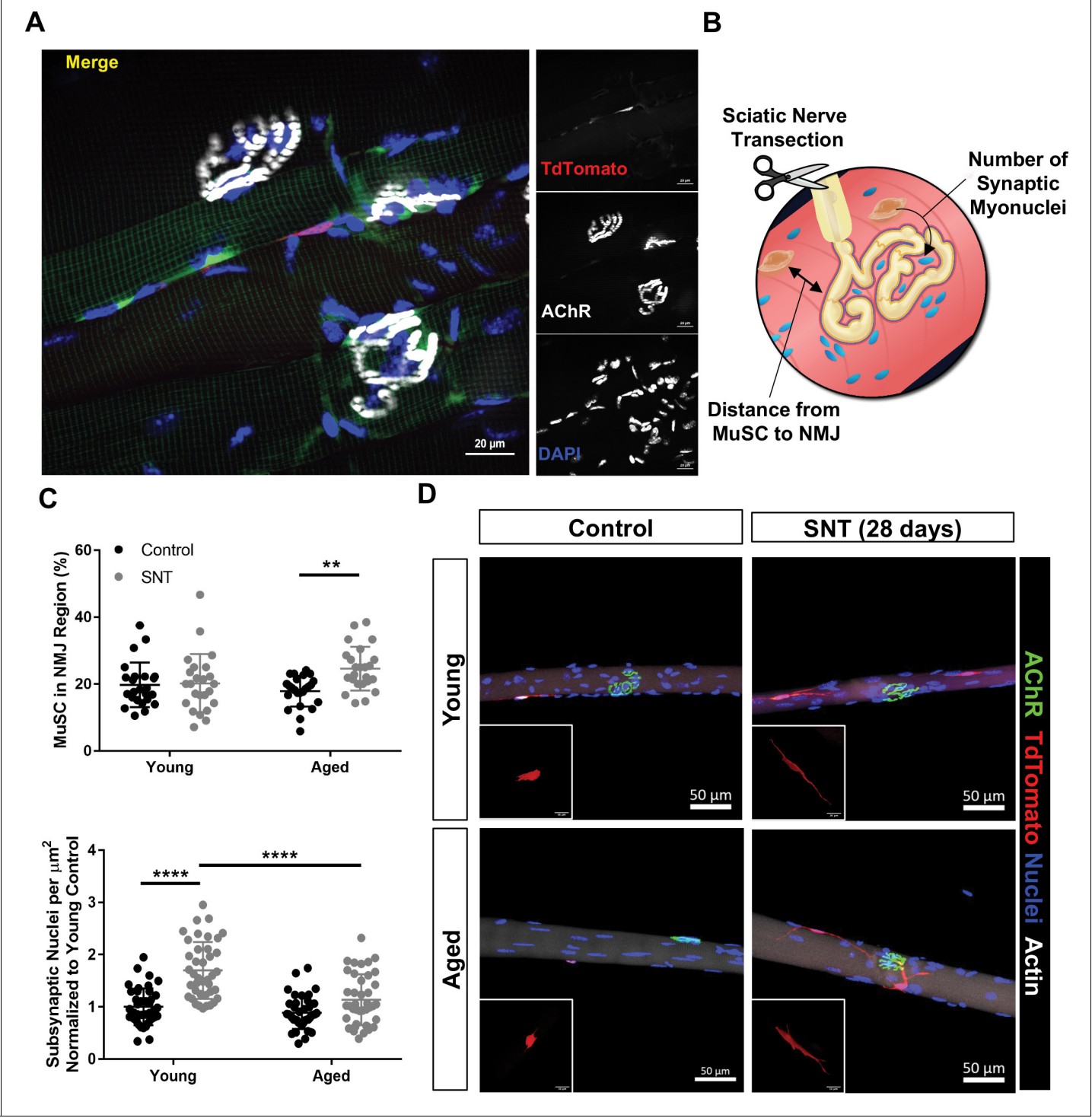

**Figure 2.** Aging attenuates muscle stem cell (MuSC) engraftment into synaptic myonuclei after nervous perturbation. (**A**) Representative immunofluorescence image of MuSC in proximity to neuromuscular junction (NMJ) from tibialis anterior (TA) muscle of young (3 months) Pax7$^{CreER/+}$-Rosa26$^{TdTomato/+}$ mouse. Pax7-Pink, acetylcholine receptor (AChR) labeled with α-bungarotoxin (BTX) stain (white) and nuclei/DAPI (blue). (**B**) Schematic of experiment to quantify distance from MuSCs to NMJ and changes in synaptic myonuclei before and after sciatic nerve transection (SNT). (**C**) Top: Quantification of the percentage of MuSCs in the NMJ region of the myofiber out of the total number of myofiber MuSCs in uninjured contralateral control and SNT muscle fibers from both aged and young mice (**p < 0.01, two-way ANOVA with Tukey post hoc test, n=4 mice/group, and >10 myofibers from each sample). Bottom: Quantification of the number of subsynaptic nuclei per synaptic area in uninjured contralateral control and SNT muscle fibers from both aged and young mice, normalized to the subsynaptic nuclei of young uninjured control (****p < 0.0001, two-way

*Figure 2 continued on next page*

Figure 2 continued

ANOVA with Tukey post hoc test). (D) Representative images of single TA uninjured (left) myofibers isolated from (top) young (2–3 months) and (bottom) aged (20–22 months) Pax7$^{CreER/+}$-Rosa26$^{TdTomato/+}$ mice. Administration of SNT and extraction of single TA myofibers 28 days post injury revealed that MuSCs are recruited to the synapse and display long projections with differential engraftment into synaptic myonuclei. Pax7/TdTomato – red, AChR/BTX – green, nuclei/DAPI – blue.

The online version of this article includes the following figure supplement(s) for figure 2:

**Figure supplement 1.** Muscle stem cell loss and changes in neuro-muscular junctions in aged muscle.

and Cre recombination, this MuSC lineage tracing system labels Pax7$^+$ MuSCs with a red fluorescent protein (TdTomato). Myofibers were isolated from the periphery of young tibialis anterior (TA) muscles (to acquire type II fibers) with the myotendinous junctions attached at each end to ensure the entire length of the fiber was isolated, and MuSCs were observed along the length of the fiber with a percentage (~20%) that were in the NMJ region (<250 µm from the NMJ, *Figure 2c*). Given that aged skeletal muscle contains degenerated NMJs, we sought to compare the abundance of MuSCs in proximity to neuromuscular synapses of young (3 months) and aged (19–22 months) mice. We utilized aged Pax7$^{CreER/+}$-Rosa26$^{TdTomato/+}$ TA muscles, administered tamoxifen as above, and observed a 30% decrease in the total number of MuSCs per myofiber in aged muscle (*Figure 2—figure supplement 1a*). Despite fewer total MuSCs per myofiber in aged muscle, the percentage of MuSCs (~18%) proximal to the NMJs was similar in young and aged muscles (*Figure 2c–d*) and young NMJs contained approximately the same number of SyM as aged NMJs (*Figure 2—figure supplement 1b*), when normalized to the total area of the endplate, which was slightly larger for aged muscles. These results suggest that despite reductions in the total number of MuSCs with aging, the subset of MuSCs proximal to the NMJ is maintained.

## Aging alters localized MuSC behavior after denervation

To test if remaining MuSCs in aged muscle displayed variations in behavior after neuromuscular disruption, we administered SNT as above and profiled MuSCs and NMJs on myofibers and myobundles 28 days post injury (dpi). After SNT, the post-synaptic nicotinic acetylcholine receptors (nAChRs) were observed to fragment and exhibit diffuse staining for young and aged myofibers. In this set of experiments, we administered tamoxifen 14 days after SNT to be able to discern MuSCs from the underlying fiber and if fusion occurred in the later stages of re-innervation. Both young and aged MuSCs converged toward the NMJ post SNT (*Figure 2c–d*) and developed long projections, with a statistically significant increase in the number and volume of MuSCs for aged muscle that were in proximity to NMJs (*Figure 2c–d*, *Figure 2—figure supplement 1*). Contrasting the number of SyM 28 days after SNT revealed a statistically significant increase in the number of SyM for young muscle in response to the injury, but no change in the number of SyM post SNT for aged muscle (*Figure 2c–d*). These results are consistent with previous reports whereby MuSC depletion leads to loss of myonuclei proximal to young NMJs after SNT (*Liu et al., 2015*). Coupling these results together suggest that MuSCs are recruited to the NMJ as a result of acute denervation, but in aging, MuSCs do not contribute SyM as efficiently as in youth.

## Identification of an NMJ-associated subset of MuSCs that increases with age

To further understand how MuSCs change with age, MuSCs were extracted from lower hind limb muscles (TA, EDL, and gastrocnemius) of young (2–3 months) and aged (22–26 months) wild-type (WT) mice using fluorescent activated cell sorting (FACS) (*Cerletti et al., 2008*), with both negative (Sca-1⁻, CD31⁻, CD45⁻, Mac-1⁻, Ter-119⁻) and positive surface markers (*Aguilar et al., 2016*) (CXCR4$^+$ and β1-integrin$^+$, *Figure 3a–b*, *Figure 3—figure supplement 1a*). Both young and aged WT FACS-sorted MuSCs (FSMs) from uninjured hind limb muscles were subjected to droplet-based single cell mRNA sequencing (scRNA-Seq) (*Zheng et al., 2017*) and a total of 34,930 single MuSC libraries were generated (12,057 young FSMs, 22,873 aged FSMs), yielding a mean of 5918 unique molecular identifiers (UMIs) and 1749 genes per cell after basic quality filtering. To further assess the purity of the FSMs, scRNA-Seq profiles of cells isolated from limb muscles across lifespan (3 months – *Tabula Muris* [*Tabula Muris Consortium, 2018*] and 24 months – *Tabula Muris Senis (TMS)* [*Tabula Muris*

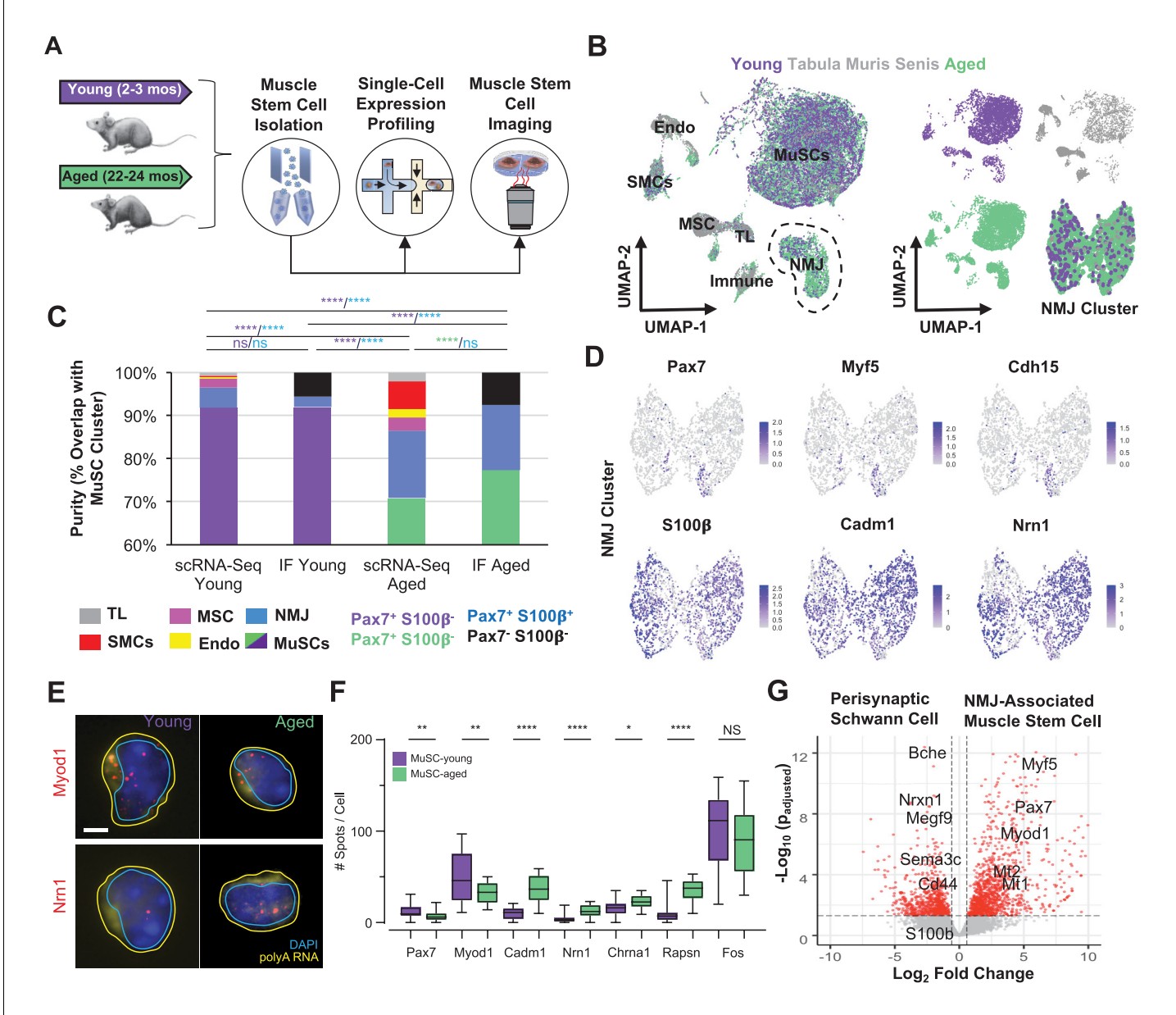

**Figure 3.** Single cell analysis of skeletal muscle stem cells (MuSCs) in age shows a subset express transcripts associated with the neuromuscular junction. (**A**) Experiment design schematic. MuSCs were fluorescent activated cell sorting (FACS) enriched from young and aged wild-type (WT) mice, then analyzed by IF, single cell mRNA sequencing (scRNA-Seq), or single-molecule fluorescence in situ hybridization (smFISH). (**B**) Left: Dimensional reduction and unsupervised clustering of young (2–3 months) and aged (22–24 months) WT FACS-isolated MuSCs and limb muscle cells from 24-month Tabula Muris Senis datasets colored by sample type showing overlap between FACS-enriched MuSCs and MuSC clusters among all mono-nucleated cells. Four and five independent replicates of young and age, respectively, were sequenced where each sample is composed of a pool from two mice. Cell types were identified according to markergene expression and labeled on the plot. NMJ – neuromuscular junction; TL – tenocyte; MSC – mesenchymal stromal cell; Immune – CD45+ immune cell; Endo – endothelial cell; SMC – smooth muscle cell. Right: Plots split by sample type showing contribution to uniform manifold approximation and projection (UMAP) and re-clustered UMAP of the NMJ-associated cluster. (**C**) Stacked bar plot representing the fraction of the sorted cell populations that clustered with each mono-nucleated cell type as well as the number of Pax7+S100b- (MuSC) and Pax7+S100b+ (NMJ) cells based on IF stains. A higher fraction of FACS purified cells from aged mice are associated with the NMJ cluster as compared to young. Quantification by FACS imaging of S100β and Pax7 proteins shows that less than 5% of young MuSCs co-express both markers compared to 15% of aged MuSCs. **** denotes $p < 0.0001$ and ns denotes $p > 0.05$ by two-sided z-test for proportions with pairwise comparisons using Holm p-value adjustment. Threshold for significance was considered $p < 0.05$. Sample sizes (n) were 3568, 483, 6532, and 880 cells for scRNA-Seq young, FACS young, scRNA-Seq aged, and FACS aged, respectively. Representative IF images are shown in *Figure 3—figure supplement 2*. (**D**) Re-clustered UMAP diagrams of the NMJ-associated cluster and overlay of specific genes. (**E**) Representative pseudocolored images of MuSCs from young

*Figure 3 continued on next page*

*Figure 3 continued*

or aged mice, stained with DAPI (blue), probes against Myod1 mRNA (red, top panels), probes against Nrn1 mRNA (red, bottom panels), and probes against polyA RNA (yellow). Nuclear (cyan) and cell (yellow) boundaries are also represented. Scale bar, 2 μm. (F) Box plots of spots/cell, wherein the central line, box edges and bars represent the median, 25th and 75th quartiles, and data range respectively. p-Values are based on two-tailed t-test with Welch's correction, ****p < 0.0001, **p<0.01, *p<0.1, NS – not significant, n=210 cells for young and n=210 cells for aged. (G) Volcano plot of differential gene expression between Pax7+ cells from the NMJ cluster and bulk RNA-Seq datasets from Schwann cells. Gene counts of the single cell datasets were merged across cells to generate 'sudo-bulk' expression matrices, where half of the cells from each sample were summed separately and considered a biological and technical replicate in order to achieve sufficient sample size. DESeq2 was used to calculate differential expression. More than 2500 genes were differentially expressed. n = 4 for Pax7+NMJ sequencing replicates and n = 5 for Schwann cell sequencing replicates. The online version of this article includes the following figure supplement(s) for figure 3:

**Figure supplement 1.** Single-cell sequencing of sorted muscle stem cells in young and aged muscle.

**Figure supplement 2.** Synaptic factors in aged muscle stem cells.

---

*Consortium and Almanzar, 2020*]) were integrated and compared (*Figure 3b*; 2301 *Tabula Muris* and 11,895 *TMS* cells, mean 4418 UMIs per cell). After filtering, data integration, and non-linear dimension reduction through uniform manifold approximation and projection (UMAP) clustering (*Becht et al., 2019*), seven clusters/cell type were observed (*Figure 3—figure supplement 1b–c*) with a clear MuSC cluster that expressed myogenic regulatory factors (*Pax7, Myf5, Myod1, Figure 3—figure supplement 1c*).

Quantification of the number of cells that clustered with MuSCs compared with other clusters revealed that ~92% of young FSMs overlapped with the MuSC cluster (*Figure 3c*). In contrast, ~73% of aged FSMs overlapped with the MuSC cluster (*Figure 3c*), and annotation of the clusters based on their unique gene expression profiles (*Figure 3—figure supplement 1c–d*) revealed that the largest non-MuSC cluster of FSMs expressed genes associated with the NMJ. Re-clustering of cells in the NMJ-associated cluster showed a subset of $Pax7^+$ cells within this cluster, which also expressed *Myf5*, a myogenic transcription factor, and *M-cadherin* (*Cdh15, Figure 3d*). This subset of cells also expressed Rapsyn (*Rapsn*), a post-synaptically expressed protein that is essential for acetylcholine receptor (AChR, *Chrna1*) clustering and formation of NMJs (*Aare et al., 2016; Figure 3—figure supplement 2a*). Other unique genes observed in the NMJ cluster were *S100b*, a $Ca^{2+}$ binding neurotrophic factor that induces neurite growth (*Sorci et al., 2013*), *Cadm1* (*Tanabe et al., 2013*), a homophilic cell adhesion molecule that promotes synapse formation (*Biederer et al., 2002*) and interacts with synaptic proteins (*Goda and Davis, 2003*), and *Neuritin 1* (*Nrn1*), a glycosylphosphatidylinositol anchored protein that is induced by neurotrophins (*Naeve et al., 1997*) and has previously been observed in MuSCs (*Seale et al., 2004*). To rule out if FACS was enriching other cells types found at NMJs such as Schwann cells or glia along with MuSCs, single young and aged FSMs were co-stained for Pax7 and S100β (*Figure 3—figure supplement 2b*). Immunostaining of young and aged FSMs with Pax7 showed that ~94% and ~93% were $Pax7^+$, respectively (*Figure 3c*, labeled 'IF Young' and 'IF Aged'). Young FSMs displayed a small fraction of $Pax7^+/S100β^+$ cells (~2%), which contrasted with the larger fraction of $Pax7^+/S100β^+$ (~16%) observed in aged FSMs, and these results were approximately consistent with the percentages identified through scRNA-Seq. Comparison of $Pax7^+$ cells with $Pax7^-$ cells in the NMJ cluster showed unique and shared expression profiles suggesting two different cell types (*Figure 3—figure supplement 2c*). To further validate young and aged FSMs differentially expressed transcripts associated with the NMJ, we profiled a subset of transcripts using single-molecule RNA in situ fluorescence hybridization (*Figure 3e*). We observed a decrease in the number of *Pax7* and *Myod1* transcripts in aged MuSCs and an increase in the number of *Cadm1, Nrn1, Rapsn* transcripts when compared to young MuSCs (*Figure 3f*) in strong concordance with our sequencing data (R=0.88, *Figure 3—figure supplement 2d*).

To further distinguish the sub-population of MuSCs from perisynaptic Schwann cells, the expression of the NMJ-associated MuSCs was contrasted with a pure population of perisynaptic Schwann cells *Castro et al., 2020*; 2576 genes were observed to be differentially expressed (*Figure 3g*, p_adj. < 0.05), and many genes upregulated in the NMJ-associated MuSCs were associated with inflammation (*Mt1, Mt2, Tnfrsf12a*, and *Crlf1*) and myogenic progenitors (*Pax7, Myod1, Myf5*), whereas upregulated genes in perisynaptic Schwann cells were associated with the synapse (*Bche, Sema3c, Megf9*) and perisynaptic Schwann cells surface receptors (*Cd44*). Overall, these results

show a subset of MuSCs that express transcripts associated with the NMJ increase in aging and are distinct from perisynaptic Schwann cells.

## Muscle injury induces a continuum of regenerative states in MuSCs that is distinct from synaptic fate induced through neural degeneration

To determine if acute muscle injury induced a similar fate to that observed in the subset of MuSCs that expressed NMJ transcripts in aged muscle, hind limb muscles of aged mice were injured via BaCl$_2$ injection (*Figure 4a*). MuSCs were purified with FACS at multiple time points (3 days, 7 dpi) and subjected to scRNA-Seq, which recovered 14,329 cells after filtering (7595 from 3 dpi and 6734 from 7 dpi, *Figure 4—figure supplement 1a–b*). We integrated datasets of single myonuclei (*Petrany et al., 2020*) RNA-sequencing from age-matched (24 months) muscles to further determine if the NMJ-associated MuSCs clustered with SyM or other myonuclei within myofibers (*Figure 4a*). UMAP dimension reduction and clustering revealed a continuum of MuSC states from uninjured to 3 dpi with activated cells from 7 dpi spanning an intermediate space (*Figure 4b*). A decrease in the percentage of FSMs that overlapped with the NMJ-associated cluster was observed for 3 and 7 dpi, respectively (*Figure 4—figure supplement 1b*). As expected, both the SyM and NMJ-associated MuSCs clustered together, and despite the MuSC heterogeneity before and after muscle injury, separation from the NMJ-associated cluster was observed for each of the time points. These results suggest that variations in MuSC fate (*Figure 4—figure supplement 1c*), and that the decision to acquire a synaptic fate may occur over longer periods than 7 days, which is consistent with the time to take to stabilize the neuromuscular synapse during development (~1 month) (*Sanes and Lichtman, 1999*). To probe further into drivers of the variance between clusters during regeneration, MuSCs and myonuclei were ordered using RNA velocity (*La Manno et al., 2018*; *Figure 4b*). scVelo (*Bergen et al., 2020*)-derived streams displayed trajectories from committed progenitors (MyoD1$^{hi}$) isolated primarily at 3 dpi toward differentiating myoblasts (MyoG$^{hi}$) that progressed toward myonuclei (Myh1$^{hi}$ or Myh4$^{hi}$) or back toward the initial uninjured state (*Figure 4c*, *Figure 4—figure supplement 1c*). These results were consistent with expression of canonical myogenic regulatory factors (*Pax7*, *Myf5*, *Myod1*, *Myog*) and previously observed regenerative dynamics5 (*Dell'Orso et al., 2019*), (*Kimmel et al., 2020*; *Figure 4c*).

## Motor neuron degeneration induces NMJ-associated MuSCs similar to aging

We next asked if an orthogonal model of neurodegeneration that triggers bouts of NMJ denervation and re-innervation and contains high levels of oxidative damage (*Jang and Van Remmen, 2011*) would display subsets of MuSCs that express NMJ transcripts and changes in MuSC fusion as observed in aging. SOD1 is an antioxidant enzyme that scavenges superoxide (O$_2$•$^-$) radicals into hydrogen peroxide (H$_2$O$_2$) and protects from oxidative damage, and *Sod1$^{-/-}$* mice (where *Sod1* is globally knocked out) display age-accelerated NMJ degeneration and muscle loss early in life (*Figure 5a*; *Jang et al., 2010*). We administered SNT to *Sod1$^{-/-}$* mice (10–15 months), extracted uninjured and denervated myofibers/fiber bundles from hind limb muscles 28 days post SNT as above. We observed degenerated NMJs in uninjured myofibers with fragmented AChRs (*Figure 5b*) and a decrease in the number of SyM after SNT (*Figure 5c*). This result is consistent with an attenuation of MuSC ability to contribute SyM, as observed in aging. To glean further insights into molecular determinants that restrict MuSC fusion into SyM, MuSCs were isolated with FACS and profiled with scRNA-Seq recovering 6609 cells. Similar to aged MuSCs, *Sod1$^{-/-}$* FSMs contained fractions of cells that did not cluster with the primary MuSC cluster (*Figure 5d*, *Figure 5—figure supplement 1a–e*). The largest non-MuSC cluster contained genes associated with NMJ function such as *S100b* (*Figure 5e*), which was consistent with results from aged muscle. IF imaging of Pax7 and S100β in *Sod1$^{-/-}$* FSMs revealed that ~97% were Pax7$^+$ (*Figure 5e*, *Figure 5—figure supplement 2a*), ruling out contamination of other NMJ cell types through FACS. *Sod1$^{-/-}$* FSMs also displayed increases in expression of a series of genes that have been shown to perturb stem cell differentiation and become dysregulated with aging (*Oh et al., 2014*) such as *Igfbp5* (*Soriano-Arroquia et al., 2016*) and *Ubb*, a critical protein in the ubiquitin-proteasome pathway (*Ryu et al., 2014*; *Figure 5f*). Integrating these data shows that degeneration of motor neurons and increases in oxidative stress from

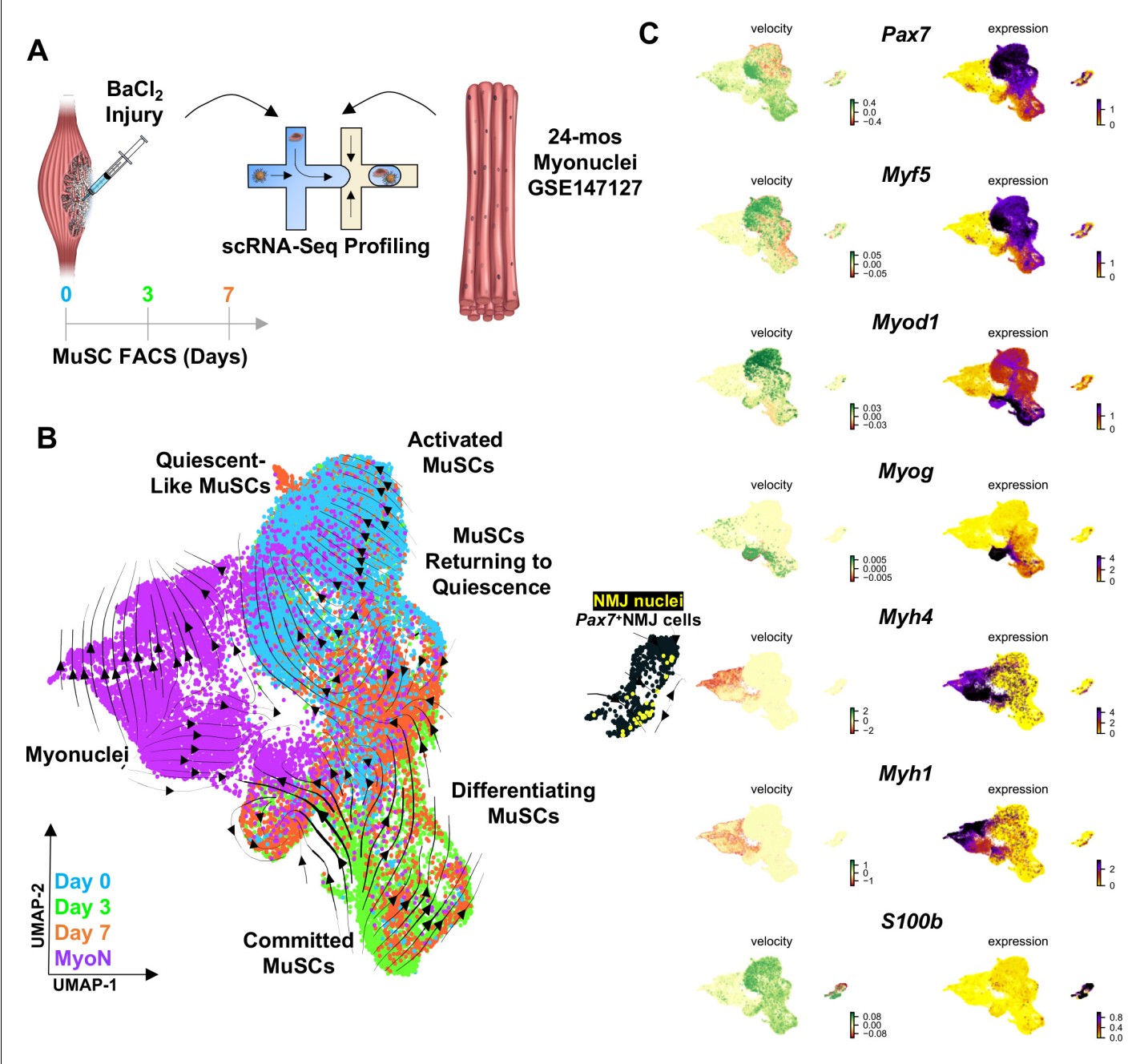

**Figure 4.** Muscle injury induces a continuum of muscle stem cell (MuSC) states distinct from the synaptic state induced by neurodegeneration. (**A**) Schematic of experiment design, whereby tibialis anterior (TA) and extensor digitorum longus (EDL) muscles from aged (22–24 months) wild-type mice were injured via intramuscular injections of barium chloride. Single cell mRNA sequencing (ScRNA-Seq) was used to profile fluorescent activated cell sorting (FACS)-enriched MuSCs before injury (day 0), and 3 and 7 days post injury (dpi). (**B**) Uniform manifold approximation and projection (UMAP) dimensional reduction of FACS-isolated MuSCs (FSMs) from injured aged muscles and uninjured, age-matched myonuclei from ref 28 colored by sample type. RNA velocity streams are overlaid onto UMAP showing MuSCs progressing from a Pax7[hi]Myf5[hi] quiescent-like state to an activated state at 0 dpi, becoming committed progenitors (Myod1[hi]) by 3dpi, and differentiating or returning to quiescence by 7dpi. (**C**) mRNA expression (right) and velocity (left) overlays of canonical MuSC (Pax7, Myf5, Myod1, and Myog), myonuclei (Myh1, Myh4), and neuromuscular junction (NMJ) (S100b) genes. The online version of this article includes the following figure supplement(s) for figure 4:

**Figure supplement 1.** Single-cell RNA sequencing of aged muscle stem cells during muscle regeneration.

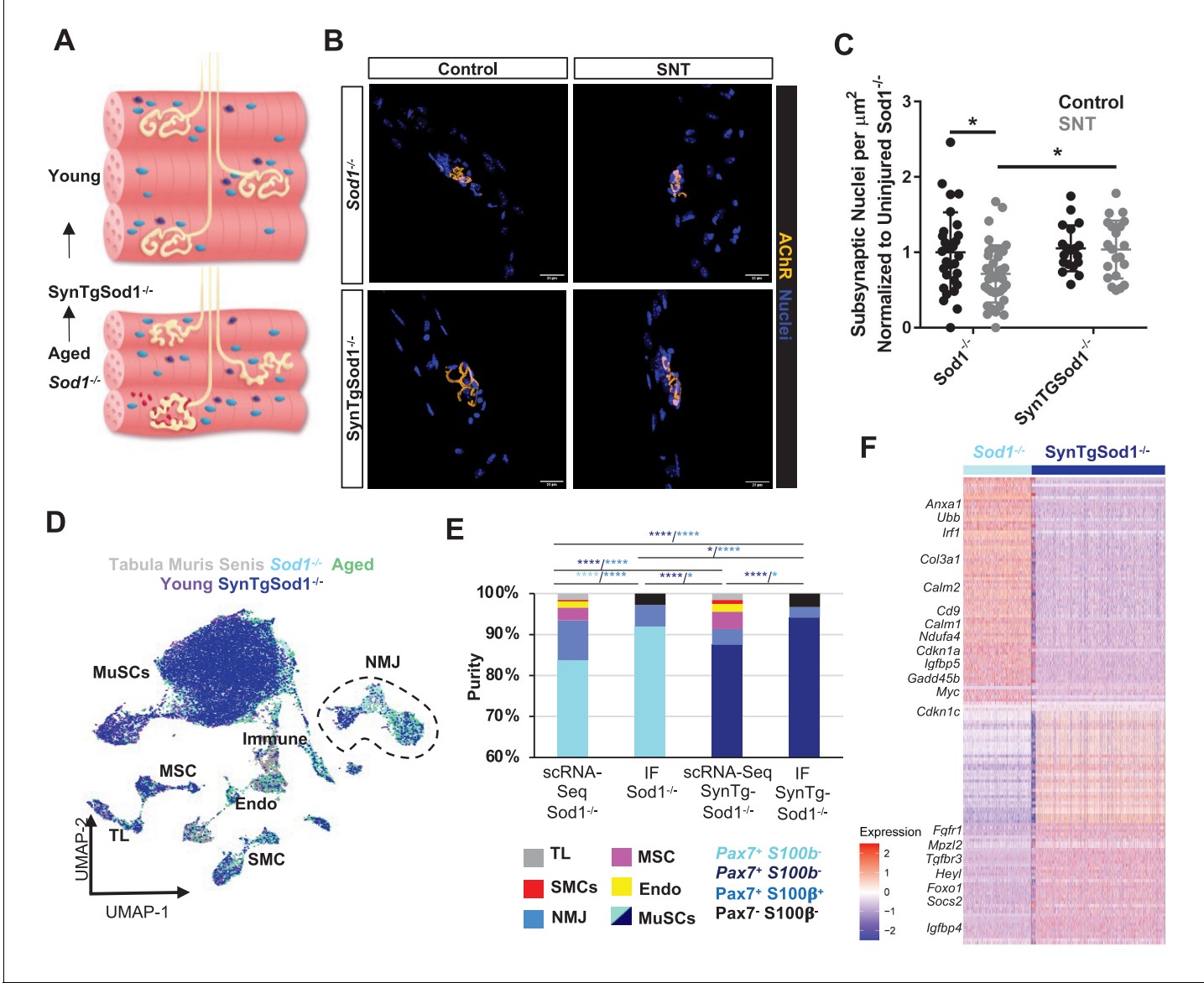

**Figure 5.** Neurodegeneration activates muscle stem cells (MuSCs) in a similar manner to aging and rescue of motor neurons partially reverses this activation state. (**A**) Schematic of neuromuscular junction (NMJ) in young (top) and aged or Sod1$^{-/-}$ (bottom) muscle, whereby NMJ becomes fragmented or partially denervated. (**B**) Representative images of Sod$^{-/-}$ and SynTgSod1$^{-/-}$ myofibers from uninjured and sciatic nerve transection (SNT) tibialis anterior muscles stained with DAPI (blue) and α-bungarotoxin (orange). (**C**) Quantification of the number of subsynaptic nuclei per synaptic area in uninjured control and SNT muscle fibers from both Sod1$^{-/-}$ and SynTgSod1$^{-/-}$ mice, normalized to the subsynaptic nuclei per area of Sod1$^{-/-}$ uninjured control (*p < 0.05, two-way ANOVA with Tukey post hoc test). (**D**) Uniform manifold approximation and projection (UMAP) dimensional reduction of fluorescent activated cell sorting (FACS)-isolated MuSCs (FSMs) from young and aged wild-type (WT) mice, 10-month Sod1$^{-/-}$ mice, 10-month Sod1$^{-/-}$ rescue (SynTgSod1$^{-/-}$) mice, and public Tabula Muris Senis limb muscle datasets colored by sample type. One and two independent replicates of Sod1$^{-/-}$ and Sod1$^{-/-}$ rescue (SynTgSod1$^{-/-}$) mice, respectively, were sequenced where each sample is composed of a pool from two mice. Cell types were identified according to markergene expression and labeled on the plot. TL – tenocyte; MSC – mesenchymal stromal cell; Immune – CD45+ immune cell; Endo – endothelial cell; SMC – smooth muscle cell. (**E**) Stacked bar plot representing the fraction of the sorted cell populations that clustered with each mono-nucleated cell type as well as the number of sorted cells that were Pax7+S100b- (MuSC) and Pax7+S100b+ (NMJ) cells based on IF staining. More FACS purified MuSCs from Sod1$^{-/-}$ mice cluster with NMJ-associated cells when compared to SynTgSod1$^{-/-}$ mice. **** denotes p < 0.0001 and * denotes p < 0.05 by two-sided z-test for proportions with pairwise comparisons using Holm p-value adjustment. Threshold for significance was considered p < 0.05. Sample sizes (n) were 6042, 1622, 10445, and 1614 cells for single cell mRNA sequencing (scRNA-Seq) Sod1$^{-/-}$, FACS Sod1$^{-/-}$, scRNA-Seq SynTgSod1$^{-/-}$, and FACS SynTgSod1$^{-/-}$, respectively. Representative IF images are shown in *Figure 5—figure supplement 2*. (**F**) Heatmap of the top 75 differentially expressed genes between MuSCs isolated from Sod1$^{-/-}$ and SynTgSod1$^{-/-}$ mice. All genes shown have a log fold change greater than 0.25 and a p_adjusted value less than 10$^{-20}$.

*Figure 5 continued on next page*

*Figure 5 continued*

The online version of this article includes the following figure supplement(s) for figure 5:

**Figure supplement 1.** Single-cell RNA sequencing of Sod1⁻/⁻, and SynTgSod1⁻/ muscle stem cells.

**Figure supplement 2.** Validation of synaptic proteins in sorted muscle stem cells from Sod1⁻/⁻ and SynTgSod1⁻/⁻ muscles.

*Sod1⁻/⁻* results in modulation of MuSCs in a similar manner to aging, whereby MuSCs do not fuse as efficiently after nerve injury and remain in muscle expressing NMJ-associated transcripts.

## Rescue of degenerative motor neurons partially reverts MuSC dysfunction

To determine if genetic rescue of motor neurons would revert the MuSC response observed in *Sod1⁻/⁻* to be more similar to that observed in young muscles, human *Sod1* was specifically overexpressed only in motor neurons via the synapsin 1 promoter (*Sakellariou et al., 2014*) (SynTgSod1⁻/⁻). Previously, we showed that this rescue prevents NMJ degeneration and muscle fiber denervation that is observed in *Sod1⁻/⁻* mice (*Su et al., 2021*; *Figure 5a*). We extracted single muscle fibers from uninjured hind limb muscles and muscles 28 days post SNT of age-matched *Sod1⁻/⁻* and Syn-TgSod1⁻/⁻ mice and profiled NMJ morphology. Prior to SNT, AChRs in *Sod1⁻/⁻* myofibers were fragmented and contained fewer SyM than myofibers of SynTgSod1⁻/⁻ mice that also displayed canonical pretzel-like clusters. After SNT, the number of SyM was increased for SynTgSod1⁻/⁻ compared to *Sod1⁻/⁻*, but not statistically different from uninjured controls (*Figure 5c*). These results suggest that restoration of the NMJ partially augments the ability of MuSCs to contribute SyM.

To understand molecular changes in MuSCs from genetic rescue of motor neurons that may contribute to changes in MuSC ability to fuse into SyM, hind limb muscles from age-matched *Sod1⁻/⁻* and SynTgSod1⁻/⁻ mice were harvested. MuSCs were extracted with FACS as above and profiled with scRNA-Seq, recovering 11,790 cells. Similar to young FSMs, SynTgSod1⁻/⁻ FSMs revealed a decrease in the percentage of FSMs that clustered with other cell types (*Figure 5d–e*) when compared to *Sod1⁻/⁻* FSMs. IF imaging of Pax7 and S100β in SynTgSod1⁻/⁻ FSMs revealed that ~96% were Pax7⁺ and displayed a comparable fraction of Pax7⁺/S100β⁺ cells as in youth (2%) (*Figure 5e*). Comparing the gene expression profiles of SynTgSod1⁻/⁻ FSMs to *Sod1⁻/⁻* FSMs revealed increases in expression of *Heyl*, *Fgfr1*, and *Foxo1*, which have been associated with quiescence and healthy MuSCs (*Figure 5f*). Amalgamating these results shows that prevention of NMJ degeneration attenuates the subset of MuSCs expressing transcripts associated with an NMJ and partially rescues MuSC ability to contribute new SyM.

## Mislocalization of S100β signaling from denervation promotes MuSC maladaptation

We reasoned that MuSCs that do not engraft into SyM after nerve injury and display increased expression of NMJ-associated transcripts may occur through chronic exposure to signaling that is normally restricted to the synapse such as from perisynaptic Schwann cells. To evaluate this, we utilized a young (6 months) murine model that contains a fluorescent reporter for S100β (S100β-GFP, *Figure 6a*) and harvested uninjured myofibers/myobundles from TA muscles (*Figure 6b*). Consistent with our results from young mice, Pax7⁺/S100β ⁺ cells were rarely observed in uninjured S100β-GFP muscle (*Figure 6c*), and very few S100β-GFP cells were detected outside of the synapse (*Figure 6c*). Next, we performed SNT on S100β-GFP muscles and extracted myofibers/myobundles 28 days after injury. As expected, we observed S100β-GFP signals outside of the synapse and a small percentage of Pax7⁺ / S100β⁺ cells in proximity to NMJs (*Figure 6b–c*). The increase in the number of S100β⁺ signals outside of nerve terminals was consistent with the known role of Schwann cells to guide regenerating axons (*Barik et al., 2016*), (*Son and Thompson, 1995*) and suggests that changes in localized cellular signaling resulting from denervation may drive MuSCs to upregulate synaptic genes such as *S100β*. To gain a deeper understanding of the functional consequences of *S100β* increases in expression in myogenic progenitors, we utilized a CRISPR transcriptional activation system (dCas9-SunTag) and transfected short guide RNAs (sgRNAs) targeted to the *S100β* locus in primary myoblasts (*Figure 6d*). We confirmed increased S100β expression after introduction of sgRNAs compared to blank controls (*Figure 6e*, *Figure 6—figure supplement 1a*), and consistent with previous

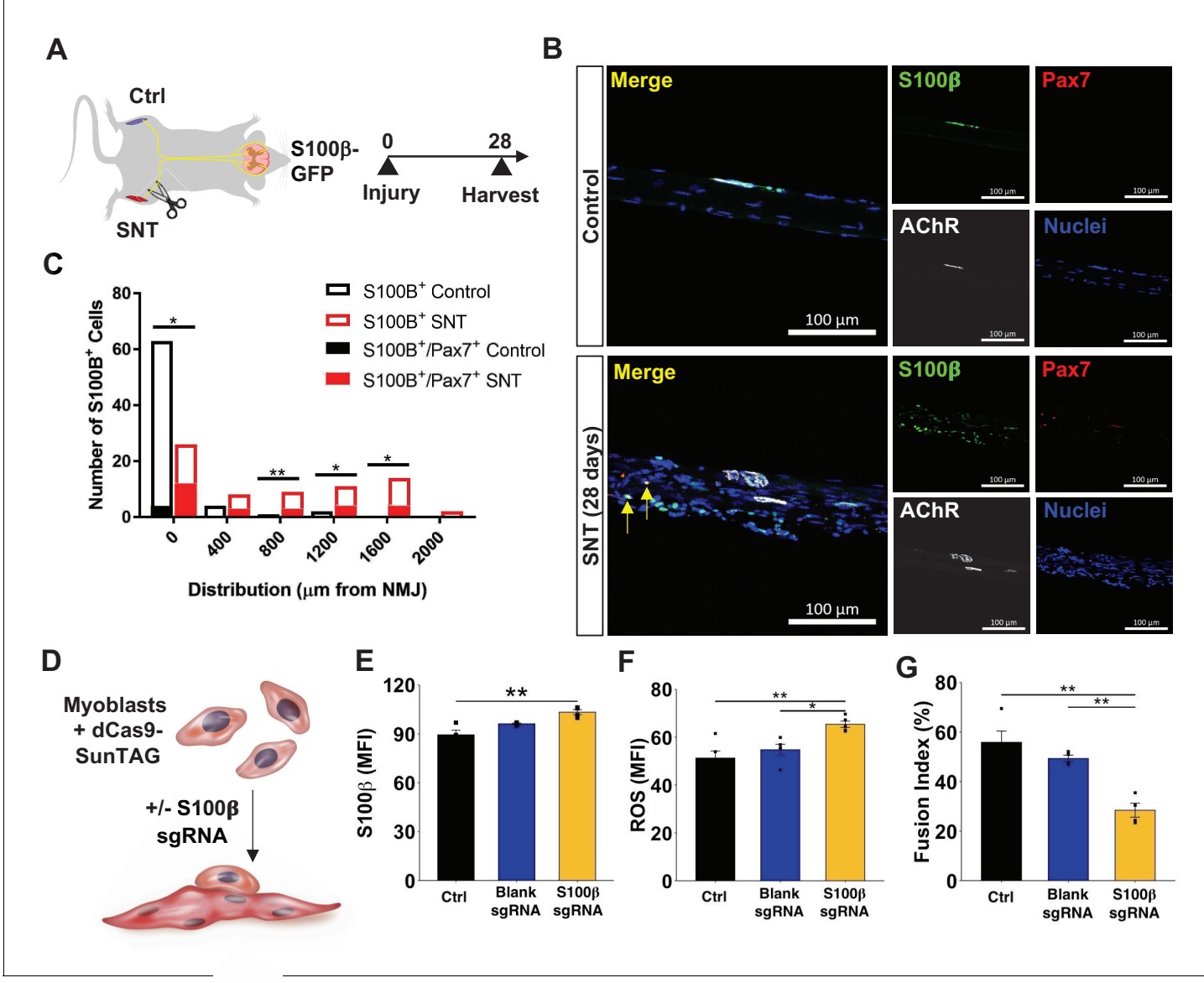

**Figure 6.** Mislocalization of S100β signaling from denervation promotes muscle stem cell maladaptation. (**A**) Schematic of S100β-GFP mice administered sciatic nerve transection (SNT) on one limb and the other limb acts as a control (Ctrl). (**B**) Representative immunofluorescence z-stack images of isolated muscle fibers from young uninjured and S100β-GFP transgenic mice administered SNT. Myofibers/myobundles are stained with Hoechst33342 (blue), Pax7 (red), α-bungarotoxin (white), and S100β-GFP (green). (**C**) Euclidean distance of S100B[+] and S100B[+]/Pax7[+] cells from the neuromuscular junction (NMJ) in tibialis anterior (TA) muscles from uninjured controls and SNT muscles. n=70 S100B[+] cells were quantified on single myofibers from four different mice. *p<0.05, **p<0.01 between groups for S100B[+] cells using two-tailed t-test with Holm multiple testing corrections. No statistical significance between groups for S100B[+]/Pax7[+] cells. (**D**) Schematic of transcriptional activation system in primary myoblasts using CRISPR-dCas9-Suntag and short guide RNA (sgRNA) targeted to S100β. (**E**) Quantification of mean fluorescent intensity (MFI) of S100β immunofluorescent staining in untransfected cells (Ctrl) and cells transfected with an empty vector (blank sgRNA) or an S100β sgRNA-containing plasmid. Sample sizes were four culture wells, each with >100 cells. **p < 0.01 by one-way ANOVA and Tukey post hoc test. (**F**) Quantification of MFI from reactive oxygen species (ROS, CellROX) for Ctrl cells and cells transfected with blank sgRNA and S100β sgRNA. Sample sizes were five culture wells, each with >100 cells. *p < 0.05 and **p < 0.01 by one-way ANOVA and Tukey post hoc test. (**G**) Quantification of fusion index for Ctrl cells and cells transfected with blank sgRNA and S100β sgRNA, then allowed to differentiate for 3 days. Fusion index was calculated as number of nuclei in myotubes/total number of nuclei. Sample size was four culture wells, each with >100 nuclei. **p < 0.01 by one-way ANOVA and Tukey post hoc test.

The online version of this article includes the following figure supplement(s) for figure 6:

**Figure supplement 1.** Overexpression of S100β induces muscle stem cell dysfunction as observed in aging.

observations (*Sorci et al., 2004*), detected increases in reactive oxygen species (ROS) (CellROX, *Figure 6F*) and apoptosis (TUNEL, *Figure 6—figure supplement 1b*). Differentiation of myogenic progenitors that overexpressed S100β displayed reductions in fusion when compared to controls, which was also in line with previous studies (*Son and Thompson, 1995*; *Figure 6g*, *Figure 6—figure supplement 1c*). These results mimic aged and $Sod1^{-/-}$ MuSCs that displayed increased S100β and reduced fusion into SyM, suggesting that denervation and chronic exposure to synaptic signals such as S100β may contribute to MuSC dysfunction observed in aging. Integrating these results permits construction of a model whereby in young healthy muscle MuSCs support the NMJ through fusion into SyM (*Figure 7*). However, in aging this support structure is attenuated and contributes to NMJ degeneration through lack of ability to promote re-innervation.

## Discussion

The neuromuscular synapse has traditionally been described to be controlled through bi-directional communication between motor neurons and SyM within muscle fibers. We show here that MuSCs integrate into this circuitry and enact a localized retrograde response after denervation by engrafting into positions near the NMJ. We also demonstrate that a subset of MuSCs are proximal to the NMJ across lifespan, and that the MuSC response to SNT is acute, but diminished with age and neurodegeneration. The inability of MuSCs to contribute SyM in aged and $Sod1^{-/-}$ muscle effectively leave behind a subset of progenitors expressing transcripts associated with the NMJ and are transcriptionally similar to SyM. These results are consistent with previous reports showing MuSCs in denervated muscle fail to terminally differentiate (*Borisov et al., 2005a*) and form smaller myotubes (*Macpherson et al., 2011*; *Borisov et al., 2005b*). We last show that the attenuated MuSC response to denervation in a $Sod1^{-/-}$ model can be improved by recovery of motor neurons. Summing these findings, we posit that MuSCs contribute to re-innervation and increases in denervation in advanced age may be in part driven or aggravated by reductions in MuSCs and inhibition to fuse into new SyM.

The focused fusion of MuSCs into NMJ proximal positions suggests denervation engenders signaling that promotes recruitment and differentiation. Impairments in this engraftment behavior of MuSCs in old age and $Sod1^{-/-}$ muscle requires further research, but our data suggests changes in

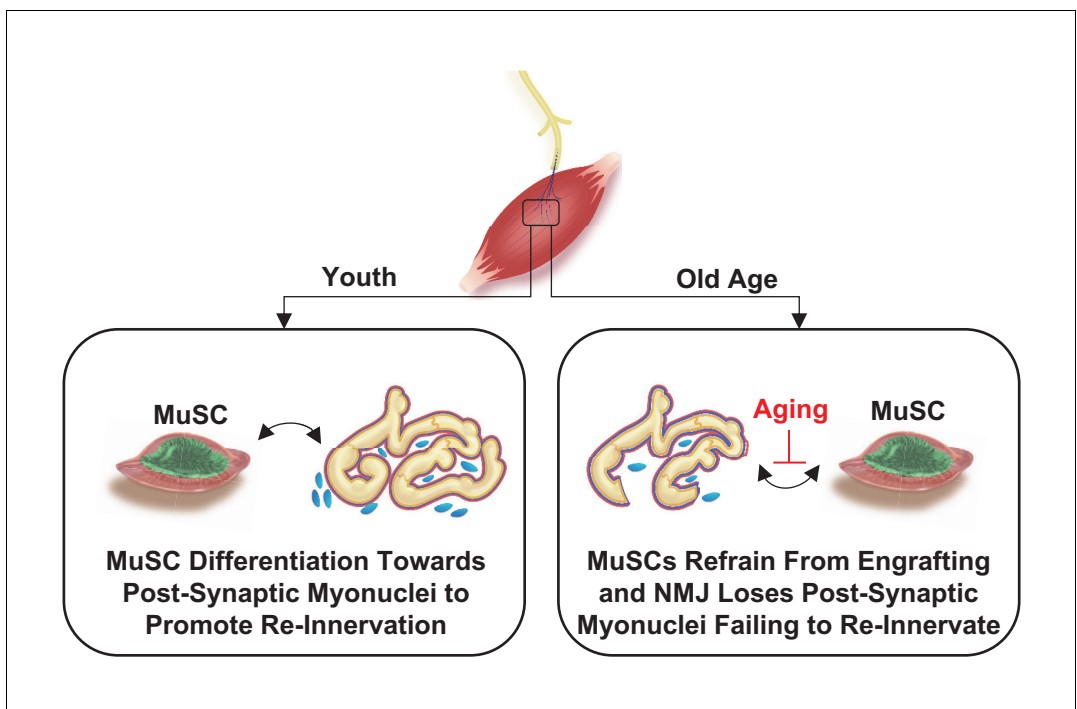

**Figure 7.** Proposed model for relationship between muscle stem cells and the neuromuscular junction.

signaling from extended exposure to signaling from Schwann cells and glia may contribute. In addition to *S100β*, Schwann cells and glia secrete *TGF-β* (*Fuentes-Medel et al., 2012*; *Feng and Ko, 2008*) after nerve injury to promote repair (*Jessen and Mirsky, 2019*), and this signaling ligand inhibits myogenic fusion (*Girardi et al., 2021*). In line with these observations, we detected increases in expression of *Igfbp5* in aged and *Sod1*-/- MuSCs compared to young and SynTgSod1-/- MuSCs, which is increased after exposure to TGF-β (*Kamanga-Sollo et al., 2005*). Conversely, Schwann cells and glia also secrete multiple neurotrophic factors (*glial cell line-derived neurotrophic factor, brain-derived neurotrophic factor, neurotrophin-3 and -4, nerve growth factor, and N-cadherin*), and these paracrine signaling factors have been shown to influence regenerative functions of MuSCs (*Mousavi and Jasmin, 2006*; *Hicks et al., 2018*) by signaling through the *ERBB3* (*Van Ho et al., 2011*) and *NGFR/p75* (*Deponti et al., 2009*) receptors. Thus, Schwann cells and glia may shift in phenotype through aging (*Painter et al., 2014*) or after persistent denervation from secretion of neurotrophic factors that positively signal to motor neurons and MuSCs (*Li et al., 2019*) to a phenotype that inhibits fusion of MuSCs through secretion of TGF-β and *S100β*. While we cannot exclude that additional factors (*Yang et al., 2021*) and other cell types may also contribute to this process, our results suggest that reductions in MuSC number and attenuation of MuSC ability to fuse in aged muscle may contribute to lack of re-innervation and degenerative changes that occur at the muscle endplate (*Li et al., 2011*), (*Valdez et al., 2010*).

Degeneration of the NMJ has been shown to be a gradual process (*Martineau et al., 2018*), resulting from imbalances between denervation and new innervation. A critical factor that prevents axonal retraction and denervation is SyM (*Grady et al., 2005*) that express AChRs and are anchored to the cytoskeleton by *Rapsn* (*Li et al., 2018*). Accordingly, the lack of engraftment from MuSCs may contribute to increased mobility of AChRs, reductions in number of SyM (*Liu et al., 2017*), and decreased ability to recruit detached axons and re-innervate (*Aare et al., 2016*; *Verdú et al., 2000*). The development of a synaptic fate in MuSCs from neural injury may be influenced at several levels. First, growing evidence (*Henriquez et al., 2008*) from in vivo and in vitro models suggests that the WNT signaling family (*Budnik and Salinas, 2011*) plays a critical role in age-associated motor unit remodeling (*Koles and Budnik, 2012*). Non-canonical WNT signaling such as from *Wnt7a* in motoneurons has been observed to enhance AChR clustering and promote neurite outgrowth (*Messéant et al., 2017*), synaptic assembly, as well as stimulate nerve terminal development (*Hall et al., 2000*) and differentiation (*Shen et al., 2018*). *Wnt7a* also confers potent effects onto MuSCs (*Le Grand et al., 2009*) and promotes engraftment as well as AChR expression. How non-canonical Wnts locally contribute to MuSC actions at the synapse and change in aging requires further study. Next, aging has been observed to induce dysregulation of calcium ($Ca^{2+}$) homeostasis, and calcium-sensitive proteins such as *S100β* were observed to be upregulated in aged and *Sod1*-/- MuSCs. The mechanism of dysregulated $Ca^{2+}$ signaling in aged MuSCs remains unresolved, but $Ca^{2+}$ influx through mechano-sensitive cation channels or voltage-gated $Ca^{2+}$ channels may play a role (*Boers et al., 2018*) and contribute to NMJ gene expression (*Chen et al., 2011*) in MuSCs. In line with this and our observations of an attenuated response of MuSCs after denervation in aging, alterations in $Ca^{2+}$ influx has been shown to modify MyoG ability to bind and activate targets (*Tang et al., 2004*). Additionally, $Ca^{2+}$-induced calmodulin-dependent protein kinase signaling has been observed to activate expression of genes associated with AChR clustering and NMJ formation such as *Rapsn* (*Li et al., 2016*). Further studies are required to elucidate if alterations in $Ca^{2+}$ homeostasis contributes to synaptic fate development in MuSC-derived myonuclei, and how these actions are modified in age.

In summary, our results provide a new insight into MuSC support of the NMJ and innervation that is lost with old age. These results suggest new opportunities for intervention through MuSCs to maintain NMJ integrity by promoting re-innervation. We envision these insights will provide further understanding of non-regenerative muscle diseases, anti-aging models, and severe forms of trauma such as volumetric muscle loss (*Aguilar et al., 2018*).

## Materials and methods

**Key resources table**

*Continued on next page*

*Continued*

| Reagent type (species) or resource | Designation | Source or reference | Identifiers | Additional information |
|---|---|---|---|---|
| Reagent type (species) or resource | Designation | Source or reference | Identifiers | Additional information |
| Antibody | APC anti-mouse Ly-6A/E (Sca-1), clone: D7, isotype: Rat IgG2a, κ | BioLegend | BioLegend 108112; RRID:AB_313349 | FACS (1:400) |
| Antibody | APC anti-mouse CD45, clone: 30-F11, isotype: Rat IgG2b, κ | BioLegend | BioLegend 103112; RRID:AB_312977 | FACS (1:400) |
| Antibody | APC anti-mouse/human CD11b, clone: M1/70. Isotype: Rat IgG2b, κ | BioLegend | BioLegend 101212; RRID:AB_312795 | FACS (1:400) |
| Antibody | APC anti-mouse TER-119, clone: TER-119, isotype: Rat IgG2b, κ | BioLegend | BioLegend 116212; RRID:AB_313713 | FACS (1:400) |
| Antibody | PE anti-mouse/rat CD29, clone: HMβ1–1, isotype: Armenian Hamster IgG | BioLegend | BioLegend 102208; RRID:AB_312885 | FACS (1:200) |
| Antibody | Biotin Rat Anti-Mouse CD184, clone: 2B11/CXCR4 (RUO), isotype: Rat IgG2b, κ, lot # 6336587 | BD Bioscience | BD Bioscience 551968; RRID:AB_394307 | FACS (1:100) |
| Antibody | Streptavidin PE-Cyanine7, lot # 4290713 | eBioscience | eBioscience 25-4317-82; RRID:AB_10116480 | FACS (1:1000) |
| Antibody | Rabbit anti-mouse laminin 1+2, Isotype: Polyclonal IgG | Abcam | Abcam ab7463; RRID:AB_305933 | IF (1:500) |
| Antibody | Goat anti-mouse IgG1, Alexa Fluor 488 conjugate | Thermo Fisher | Thermo Fisher A21121; RRID:AB_2535764 | IF (1:500) |
| Antibody | Goat anti-rabbit H+L, Alexa Fluor 555 conjugate | Thermo Fisher | Thermo Fisher A21428; RRID:AB_141784 | IF (1:500) |
| Other | 4′,6-Diamidino-2-Phenylindole, Dihydrochloride (DAPI), FluoroPure grade | Thermo Fisher | Thermo Fisher D21490 | IF (1:500) Nuclear stain |
| Antibody | Mouse anti-Pax7, Isotype MIgG1, kappa light chain, supernatant | Developmental Studies Hybridoma Bank | DHSB Pax7-s | IF (1:100) |
| Antibody | Rabbit polyclonal anti-S100B | Thermo Fisher | Thermo Fisher PA5-78161; RRID:AB_2736549 | IF (1:500) |
| Antibody | Mouse monoclonal anti-MyoD (G-1) | Santa Cruz Biotechnology | Santa Cruz Biotechnology Sc-377460; RRID:AB_2813894 | IF (1:200) |
| Antibody | Donkey anti-rabbit IgG (H+L), Alexa Fluor 647 | Thermo Fisher | Thermo Fisher A31573; RRID:AB_2536183 | IF (1:500) |
| Antibody | Goat anti-mouse H+L, Alexa Fluor 555 conjugate | Thermo Fisher | Thermo Fisher A28180; RRID:AB_2536164 | IF (1:250) |
| Antibody | Alexa Fluor 488 goat anti-rabbit secondary antibody | Thermo Fisher | Thermo Fisher A11034; RRID:AB_2576217 | IF (1:500) |
| Antibody | Alexa Fluor 488 conjugated α-Bungarotoxin | Thermo Fisher | Thermo Fisher B13422 | IF (1:500) |
| Antibody | Hoechst 33342 | Thermo Fisher | Thermo Fisher H3570 | IF (1:500) |
| Antibody | Goat anti-rabbit IgG (H+L), Alexa Fluor 647 | Thermo Fisher | Thermo Fisher A21245; RRID:AB_2535813 | IF (1:500) |
| Antibody | Goat anti-mouse IgG (H+L), Alexa Fluor 647 | Thermo Fisher | Thermo Fisher A21235; RRID:AB_2535804 | IF (1:500) |

*Continued*

| Reagent type (species) or resource | Designation | Source or reference | Identifiers | Additional information |
|---|---|---|---|---|
| Other | Dispase II (activity ≥0.5 units/mg solid) | Sigma Aldrich | Sigma D4693-1G | Digestion of tissue to extract MuSCs |
| Other | Collagenase Type II | Thermo Fisher | Thermo Fisher 17101015 | Digestion of tissue to extract MuSCs |
| Chemical compound, drug | Propidium Iodide – 1.0 mg/mL solution in water | Thermo Fisher | Thermo Fisher P3566 | FACS (1:400) |
| Commercial assay or kit | Mouse on Mouse blocking reagent | Vector Laboratories | Vector Laboratories MKB-2213; RRID:AB_2336587 | |
| Other | Corning CellTak | Fisher Scientific | Fisher Scientific C354240 | 22.4 μg/mL Adhere MuSCs to tissue culture plastic |
| Peptide, recombinant protein | XbaI | New England Biolabs | NEB R0145S | |
| Peptide, recombinant protein | SalI | New England Biolabs | NEB R3138S | |
| Peptide, recombinant protein | Matrigel Basement Membrane Matrix, LDEV-free | Corning | Corning 356234 | |
| Commercial assay or kit | CellROX Deep Red Reagent | Thermo Fisher | Thermo Fisher C10422 | |
| Commercial assay or kit | Stellaris RNA FISH Wash Buffer A | LGC-Biosearch | LGC-Biosearch SMF-WA1-60 | |
| Commercial assay or kit | Stellaris RNA FISH Wash Buffer B | LGC-Biosearch | LGC-Biosearch SMF-WB1-20 | |
| Commercial assay or kit | Stellaris RNA FISH Hybridization Buffer | LGC-Biosearch | LGC-Biosearch SMF-HB1-10 | |
| Sequence-based reagent | Fos FISH probe | LGC-Biosearch | LGC-Biosearch VSMF-3011–5 | |
| Commercial assay or kit | Single cell 3′ Library and Gel Bead Kit v2 and v3 | 10× Genomics | 10× Genomics 120267 | |
| Commercial assay or kit | QIAprep Spin Miniprep Kit | Qiagen | Qiagen 27104 | |
| Commercial assay or kit | Satellite Cell Isolation Kit, mouse | Miltenyi Biotec | Miltenyi Biotec 130104268 | |
| Commercial assay or kit | In Situ Cell Death Detection Kit, TMR red | Sigma Aldrich | Sigma 12156792910 | |
| Biological sample (*Mus musculus*) | C57BL/6 mice | Charles River Labs | C57BL/6NCrl Strain 027 | Female, 3–4 months |
| Biological sample (*Mus musculus*) | C57BL/6 mice | NIA | C57BL/6JNia; RRID:IMSR_JAX:000664 | Female, 20–24 months |
| Biological sample (*Mus musculus*) | SynTgSod1$^{-/-}$ mice | Oklahoma Medical Research Foundation | PMID:24378874, 31470261 | Female, 10–12 months |
| Biological sample (*Mus musculus*) | Sod1$^{-/-}$ mice | Oklahoma Medical Research Foundation | PMID:9264557, 15531919 | Female, 10–12 months |
| Biological sample (*Mus musculus*) | Pax7CreER$^{+/+}$ mice | Jackson Labs | RRID:IMSR_JAX:017763 | Male, 2–3 months |
| Biological sample (*Mus musculus*) | Rosa26$^{mTmG+/+}$ mice | Jackson Labs | RRID:IMSR_JAX: 007676 | Female, 2–3 months |
| Biological sample (*Mus musculus*) | Rosa26$^{TdTomato+/+}$ mice | Jackson Labs | RRID:IMSR_JAX: 007909 | Female, 2–3 months |
| Biological sample (*Mus musculus*) | Rosa26$^{nTnG+/+}$ mice | Jackson Labs | RRID:IMSR_JAX: 023035 | Female, 2–3 months |

*Continued on next page*

*Continued*

| Reagent type (species) or resource | Designation | Source or reference | Identifiers | Additional information |
|---|---|---|---|---|
| Biological sample (*Mus musculus*) | S100b-GFP mice | Jackson Labs | RRID:IMSR_JAX: 005621 | Female |
| Biological sample (*Mus musculus*) | Rosa26$^{dCas9-SunTag/+}$ mice | Jackson Labs | Stock # 43926-JAX | Female |
| Software, algorithm | CellRanger v2.0.0, v3.0.0 | 10× Genomics | https://support.10xgenomics.com/single-cell-gene-expression/software/downloads | |
| Software, algorithm | R v.4.0.2 | The R Foundation for Statistical Computing | RRID:SCR_001905 | |
| Software, algorithm | Bioconductor v.3.11 | *Huber et al., 2015* | Nature Methods 12:115–121, PMID:25633503 | |
| Software, algorithm | DESeq2 v.1.28.1 | *Love et al., 2014* | RRID:SCR_015687 | |
| Software, algorithm | scVelo v.0.2.2 | *Bergen et al., 2020* | RRID:SCR_018168 | |
| Software, algorithm | ggplot2 v.3.3.2 | *Wickham, 2016* | RRID:SCR_014601 | |
| Software, algorithm | EnhancedVolcano v.1.6.0 | *Blighe et al., 2021* | RRID:SCR_018931 | |
| Software, algorithm | Python v3.6.5 | Python Software Foundation | RRID:SCR_008394 | |
| Software, algorithm | Seurat v.3.2.2 | *Stuart et al., 2019* | RRID:SCR_007322 | |
| Software, algorithm | SeuratWrappers v0.3.0 | | https://github.com/satijalab/seurat-wrappers | |
| Software, algorithm | scCATCH | *Shao et al., 2020* | https://github.com/ZJUFanLab/scCATCH | |
| Software, algorithm | LIGER v0.5.0 | *Welch et al., 2019* | RRID:SCR_018100 | |
| Software, algorithm | AnnotationDbi v1.50.3 | *Pagès et al., 2021* | | |
| Software, algorithm | GenomeInfoDb 1.24.2 | *Arora and Morgan, 2021* | | |
| Software, algorithm | Jupyter v.4.4.0, notebook v.5.7.8 | Project Jupyter 2019 | https://jupyter.org/ | |
| Software, algorithm | dplyr v1.0.2 | *Wickham, 2021* | RRID:SCR_016708 | |
| Software, algorithm | tibble v3.4.0 | *Müller and Wickham, 2021* | | |
| Software, algorithm | tximport v1.16.1 | *Soneson et al., 2016* | RRID:SCR_016752 | |
| Software, algorithm | Velocyto v0.17 | *La Manno et al., 2018* | RRID:SCR_018167 | |
| Software, algorithm | Kallisto v0.44 | *Bray et al., 2016* | RRID:SCR_016582 | |

## Animal and injury model

C57BL/6J WT female mice were obtained from Charles River Breeding Laboratories, the National Institute on Aging, or from a breeding colony at the University of Michigan (UM). S100B-GFP mice were obtained from a breeding colony at Brown University. Two- to three-month-old Pax7$^{CreER/+}$ male mice were crossed with 2- to 3-month-old Rosa26$^{mTmG/+}$ female mice to generate Pax7$^{CreER/+}$-Rosa26$^{mTmG/+}$ and genotyped. Two- to three-month-old Pax7$^{CreER/+}$ male mice were bred with mice carrying a loxP-flanked STOP cassette followed by *TdTomato* in the ROSA26 locus and genotyped. To activate GFP expression in the Pax7$^{CreER/+}$-Rosa26$^{mTmG/+}$, Pax7$^{CreER/+}$-Rosa26$^{nTnG/+}$ mice, 4- to 6-month-old mice were administered tamoxifen (100 mg/kg body weight) through intraperitoneal injection five times (1× per day) and allowed to recover for 2–5 days. Pax7$^{CreER/+}$-Rosa26$^{TdTomato/+}$ mice were fed with tamoxifen chow for 2 weeks prior to euthanasia to induce TdTomato expression in Pax7$^+$ cells through the Cre recombinase activity and recombination efficiency was measured at >90%. All mice were fed normal chow ad libitum and housed on a 12:12 hr light-dark cycle under UM veterinary staff supervision. All procedures were approved by the University Committee on the Use and Care of Animals at UM and were in accordance with the U.S. National Institute of Health (NIH). Young female mice (4–6 months) and aged female mice (19–24 months) were randomly assigned to one of five groups: uninjured, day 3, and day 7 injured (n=4 per group). To induce skeletal muscle injury, mice were first anesthetized with 2% isoflurane and administered a 1.2% barium

chloride (BaCl$_2$) solution injected intramuscularly into several points of the TA and gastrocnemius muscles for a total of 80 µL per hind limb.

## Satellite cell isolation via FACS

For tissue collection, mice were anesthetized with 3% isoflurane, then euthanized by cervical dislocation, bilateral pneumothorax, and removal of the heart. Hind limb muscles (TA and gastrocnemius) of control and experimental mice were quickly harvested using sterile surgical tools and placed in separate plastic petri dishes containing cold phosphate-buffered saline (PBS). Using surgical scissors, muscle tissues were minced and transferred into 50 mL conical tubes containing 20 mL of digest solution (2.5 U/mL Dispase II and 0.2% [~5500 U/mL] Collagenase Type II in Dulbecco's modified Eagle medium [DMEM] per mouse). Samples were incubated on a rocker placed in a 37°C incubator for 60 min with manual pipetting the solution up and down to break up tissue every 30 min using an FBS-coated 10 mL serological pipette. Once the digestion was completed, 20 mL of F10 media containing 20% heat inactivated FBS was added into each sample to inactivate enzyme activity. The solution was then filtered through a 70 µm cell strainer into a new 50 mL conical tube and centrifuged again at 350× *g* for 5 min. The pellets were re-suspended in 6 mL of staining media (2% heat inactivated FBS in Hank's buffered salt solution) and divided into separate FACS tubes. The FACS tubes were centrifuged at 350× *g* for 5 min and supernatants discarded. The cell pellets were then re-suspended in 200 µL of staining media and antibody cocktail containing Sca-1:APC (1:400), CD45: APC (1:400), CD11b:APC (1:400), Ter119:APC (1:400), CD29/β1-integrin:PE (1:200), and CD184/ CXCR-4: BIOTIN (1:100) and incubated for 30 min on ice in the dark. Cells and antibodies were diluted in 3 mL of staining solution, centrifuged at 350× *g* for 5 min, and supernatants discarded. Pellets were re-suspended in 200 µL staining solution containing PECy7:STREPTAVIDIN (1:100) and incubated on ice for 20 min in the dark. Again, samples were diluted in 3 mL staining solution, centrifuged, supernatants discarded, and pellets re-suspended in 200 µL staining buffer. Live cells were sorted from the suspension via addition of 1 µg of propidium iodide stain into each experimental sample and all samples were filtered through 70 µm cell strainers before the FACS. Cell sorting was done using a BD FACS Aria III Cell Sorter (BD Biosciences, San Jose, CA) and APC negative, PE/ PECy7 double-positive MuSCs were sorted into staining solution for immediate processing.

## Single cell mRNA sequencing

Freshly isolated MuSCs were sorted into staining solution, enumerated by hemocytometer, and re-suspended into PBS. Cells were loaded into the 10× Genomics chromium single cell controller for each time point and age group was captured into nanoliter-scale gel bead-in-emulsions. cDNAs were prepared using the single cell 3′ Protocol as per manufacturer's instructions and sequenced on a NextSeq 500 instrument (Illumina) or NovaSeq instrument (Illumina) with 26 bases for read1 and 98 bases for read2.

## Immunostaining of NMJs

For assessment of nGFP activity and location, Pax7$^{CreER/+}$-Rosa26$^{nTnG/+}$ mice were used. Myofibers were purified by conventional collagenase digestion and trituration with fire polished glass pipettes as previously described (*Zammit et al., 2004*). Briefly, the EDL muscle was dissected, rinsed in Dulbecco's PBS, put into a 1.5 mL Eppendorf tube containing 1 mL 0.1% type I collagenase (Invitrogen) and 0.1% type II collagenase (Invitrogen) in DMEM (Sigma-Aldrich, St. Louis, MO), incubated in a shaker water bath at 37°C for 75 min and gently mixed by inversion periodically. Following digestion, the muscle was transferred to 100 mm × 15 mm plastic petri dishes containing 10 mL of plating media (10% horse serum in DMEM) using fire-polished-tip Pasteur pipettes. Under a stereo dissecting microscope, single myofibers were released by gently triturating the EDL with a series of modified Pasteur pipettes that varied in tip diameter to accommodate the progressive decrease in muscle trunk size. Inseparable fibers and debris were removed. Purified single myofibers were fixed with 4% paraformaldehyde (PFA) for 3 min, washed with PBS, and transferred to 5 mL polystyrene cell collection tubes for GFP, Pax7 and BTX immunofluorescence. The percentage of myofibers that had nGFP+ myonuclei at NMJs was enumerated. Images were acquired using a Zeiss Axio Observer Z1 or LSM 780 confocal microscope. n=5 muscles (3 males and 2 females) for each injury type, and 20–30 myofibers were counted per muscle. Routs analysis was used to identify outliers and one data

point was excluded. Two-sided two-sample Student's t-test was used, and groups were determined statistically significantly different (p = 0.0012, difference between means = 16.5925, 95% confidence interval = (9.0802, 24.1048), t = 5.2228, 7 degrees of freedom). Effect size was estimated using Cohen's d to be 3.39.

## Immunostaining of S100B, Pax7, and MyoD

Clear 8-well coverslides were coated with 22.4 µg/mL CellTak in PBS for 20 min at room temperature (RT), followed by three quick rinses with distilled water. FACS-enriched MuSCs were then seeded at a density of 10,000 cells/well and allowed to adhere for 45 min in staining buffer at RT. After adhering to the cover slide, staining buffer was aspirated and cells were fixed with 4% PFA in PBS at RT for 20 min. PFA was aspirated and after three quick washes with PBS, cells were permeabilized with 0.1% TritonX-100 and blocked with 1% BSA, 0.1% Tween-20, and 22.52 mg/mL glycine in PBS. After blocking, cells were incubated with primary antibodies (1:500 dilution of anti-S100B, 1:10 dilution of anti-Pax7, 1:200 dilution of anti-MyoD) overnight at 4°C followed by secondary antibodies (1:300 dilution of AF555 anti-mouse, 1:300 dilution of AF647 anti-rabbit) overnight at 4°C. Nuclei were stained with Hoechst 33342 (1.5 µg/mL) in PBS for 1 min at RT. Immunolabeled cells were imaged on a Zeiss epifluorescent microscope using a 10× objective. Single and double-positive cells were manually counted to calculate abundances. n=483 cells from young mice (94.4% $Pax7^+$, 2.5% $Pax7^+S100\beta^+$) and n=880 cells from aged mice (92.5% $Pax7^+$, 15.9% $Pax7^+S100\beta^+$), both sorted separately from two animals and immunostained in triplicate. Statistical comparison between groups was performed using two-sided, two-proportion z-test, where p-values less than 0.05 were considered significant. No significant difference was observed among $Pax7^+$ proportions between young and aged mice (z = 1.33, p = 0.1826), but the proportion of $Pax7^+S100\beta^+$ cells was significantly different (z = −7.56, p = 4.16×$10^{-14}$). Further, no significant difference was observed in Pax7 expression among sorted young and sorted $SOD1^{-/-}$ MuSCs (z = −0.87, p = 0.3846).

## Sciatic nerve transection

Mice were anesthetized by inhalation of 1–3% isoflurane. The hindquarter was then carefully shaved, and depilation completed with generic Nair hair removal cream. The skin was wiped three times with chlorhexidine and 70% alcohol. Then a small skin incision (<10 mm) was made 1 mm posterior and parallel to the femur, and the biceps femoris was bluntly split to expose the sciatic nerve. On the left leg, 1–2 mm sciatic nerve was then transected 5 mm proximal to its trifurcation, followed with realignment of the distal and proximal nerve ends and closure of the incision with wound clips (Autoclip, BD Clay Adams, Franklin Lakes, NJ). Sham surgery was performed on the contralateral leg where procedures were performed without nerve transection. Mice were given analgesic (0.5–1.0 mg/kg buprenorphine) and allowed to recover on a heating pad. The wound clips were removed 14 days after the SNT surgery. SNTs of S100-GFP mice were performed at Brown University. For these experiments, young adult (6 months of age), female S100β-GFP mice were deeply anesthetized with ketamine/xylazine intraperitoneal injection (10 mg xylazine and 90 mg ketamine per kg body weight). After anesthesia, a small animal trimmer and depilatory cream was used to remove the fur on one leg, the surgical site was disinfected with 70% ethanol, and a small skin incision was made. The sciatic nerve from one leg was cut near the hip and the incision closed with 4–0 nonfilament sutures. Analgesia (0.5–1 mg buprenorphine per kg body weight) was administered at time of anesthesia. The contralateral leg was left unoperated for each mouse. The TA and EDL muscles were harvested from injured and uninjured legs 28 days later for histological analyses.

## Single myofiber MuSC analysis

For nGFP localization, myofibers were purified by conventional collagenase digestion and trituration with fire polished glass pipettes as previously described (*Zhang et al., 2018*). Briefly, the EDL muscle was dissected, rinsed in Dulbecco's phosphate-buffered saline (PBS), put into a 1.5 ml Eppendorf tube containing 1 mL 0.1% type I collagenase (Invitrogen) and 0.1% type II collagenase (Invitrogen) in DMEM (Sigma-Aldrich, St. Louis, MO), incubated in a shaker water bath at 37°C for 75 min and gently mixed by inversion periodically. Following digestion, the muscle was transferred to 100 mm × 15 mm plastic petri dishes containing 10 mL of plating media (10% horse serum in DMEM) using fire-polished-tip Pasteur pipettes. Under a stereo dissecting microscope, single myofibers were

released by gently triturating the EDL with a series of modified Pasteur pipettes that varied in tip diameter to accommodate the progressive decrease in muscle trunk size. Inseparable fibers and debris were removed. Purified single myofibers were fixed with 4% PFA for 3 min, washed with PBS and transferred to 5 mL polystyrene cell collection tubes for nGFP, Pax7, and AChR immunofluorescence.

For MuSC, NMJ, and myonuclear imaging from Pax7-TdTomato mice, myofibers were isolated from the periphery of the TA muscle. TAs were first fixed in anatomical position with 4% PFA, and fiber bundles were mechanically separated for immunostaining. Antibodies used for immunostaining included Hoechst 33342 (Life Technologies H3570) for nuclei, phalloidin conjugated to Alexa Fluor 647 (Thermo Fisher Scientific A22287) for F-actin, α-bungarotoxin conjugated to Alexa Fluor 488 (Thermo Fisher Scientific B13422) for α-AChR, and S100b (Thermo Fisher Scientific PA5-78161). Following immunostaining, single myofibers were carefully mechanically separated under a dissecting stereoscope using fine forceps; 20–30 myofibers were collected from each muscle for imaging and analysis, with a biological sample size of four mice per group. Myonuclei were identified as subsynaptic myonuclei if the nucleus was in direct contact with the post-synaptic AChR endplate. MuSCs were considered near the NMJ if they were within 250 μm of the NMJ. At least 20 NMJs were assessed per muscle, with four mice per group. 3D reconstructed MuSCs were created by taking z-stack images at 1 μm intervals on a Zeiss 700 Confocal Microscope. 3D renderings were generated using Volocity and voxel intensities were quantified on ImageJ.

For Pax7 and NMJ staining from S100B-GFP mice, freshly dissected TA muscles were separated into fiber bundles along the periphery of the muscle. Fiber bundles were pinned into petri dishes containing PDMS at the myotendinous junctions and treated with 0.02% collagenase type II at 37°C for 10 min to break up connective tissue and extracellular matrix. Pinned fiber bundles were then fixed with 4% PFA at RT for 5 min, then 1% SDS at RT for 15 min and citrate buffer (10 mM sodium citrate, 0.05% Tween-20, pH 6.0) at 37°C for 1 hr in order to retrieve antigens. Fiber bundles were unpinned and placed in 1.5 mL tubes with blocking buffer for two nights, mouse IgG (1:500) for 2 hr, Pax7 antibody (1:100) for three nights, and anti-mouse secondary antibody (1:250) as well as BTX for 2 hr, with 30 min PBS washing between all staining steps. Single myofibers were mechanically isolated using forceps and placed on slides with DAPI-containing mounting media for imaging. Cells were considered S100B+ if they co-expressed DAPI and S100B, and cells that co-expressed DAPI, S100B, and Pax7 were considered S100B+/Pax7+. We measured the Euclidean distance of at least 70 S100B+ cells from the NMJ in both young control and young SNT single muscle fibers to generate a distribution of distances of S100B+ and S100B+/Pax7+ cells from the NMJ.

## Single-molecule RNA fluorescence in situ hybridization

MuSCs were enriched by FACS as described above and seeded onto 96-well glass bottom dishes (Greiner Sensoplate, Sigma, M4187-16EA) coated with 22.4 μg/mL CellTak. The plate was centrifuged at 500× g for 5 min at 4°C to allow cells to adhere to the surface. smFISH was performed as described (*Pitchiaya et al., 2019*) with minor modifications. Briefly, cells were fixed with warm (37°C) 4% PFA (prepared from 16% PFA, Thermo Fisher Scientific, 50-980-487) in 1× PBS for 10 min at 25°C, immediately after removal of growth media to avoid cell loss. Cells were washed three times with 1× PBS and permeabilized with 70% ethanol for at least 2 hr at 4°C. Cells were subsequently rehydrated in Stellaris RNA FISH Wash Buffer A (LGC-Biosearch, SMF-WA1-60) for 5 min at 25°C. Cells were subsequently treated with Stellaris RNA FISH Hybridization Buffer (LGC-Biosearch, SMF-HB1-10) containing 200 nM custom designed probes (using https://www.biosearchtech.com/stellaris-designer, coupled with Quasar 670 dye, sequence in MUSC-GEanalysis-smFISHprobes.xlsx) or 200 nM cataloged probes against Fos (VSMF-3011–5, LGC-Biosearch, coupled with Quasar 670 dye), 10 nM Oligo-dT probes (coupled with Alexa Fluor 488 dye, custom synthesis, IDT) and 10% formamide (Thermo Fisher Scientific, 15515026) for at least 4 hr at 37°C in a humidified environment. Cells were then washed once with Stellaris RNA FISH Wash Buffer A containing 10% formamide for 30 min at 37°C in a humidified environment, followed by another wash in Stellaris RNA FISH Wash Buffer A containing 20 ng/mL DAPI and 10% formamide under the same conditions. Cells were then washed with Stellaris RNA FISH Wash Buffer B (LGC-Biosearch, SMF-WB1-20) for 5 min at 25°C and mounted in buffer containing 10 mM Tris-HCl, pH 7.5, 2× SSC, 2 mM trolox, 50 μM protocatechiuc acid, and 50 nM protocatechuate dehydrogenase.

## smFISH imaging

Stained cells were imaged in three dimensions using HILO illumination at 400× magnification (100×, 1.4 NA objective coupled with a 4× magnifier) by sequential imaging using 405 nm (DAPI), 488 nm (Alexa Fluor 488) and 640 nm (Quasar 670) lasers equipped on a microscope that was previously described (*Schultz and Lipton, 1982*). Images were processed using custom-written macros in ImageJ. Analysis routines comprise three major steps: background subtraction, Laplacian of Gaussian filtering, and thresholding. Spots with intensity above set threshold are represented in images. Nuclear and cell boundary were defined by Huang thresholding DAPI and Alexa Fluor 488 images, respectively.

## Plasmid construction

An empty pCWB-U6-cloning-CMV-eGFP plasmid construct was provided by the UM Vector Core. Subsequently, a 20 bp sgRNA targeting the transcriptional start site of S100B (F: 5'- GAACA TTGGCCCAGTTCCAA-3'; R: 5'-TTGGAACTGGGCCAATGTTC-3') was ligated into the plasmid construction using XbaI and SalI-HF restriction enzymes. Plasmid DNA was purified using the QIAprep Spin Miniprep Kit, according to the manufacturer's protocol. The sequence of the construct was confirmed by DNA sequencing.

## Satellite cell isolation via MACS

Hind limb muscles from two tamoxifen-treated Pax7$^{CreER/+}$-Rosa26$^{dCas9-SunTag/+}$ mice, aged 6 months, were collected and digested as described above. Satellite cells from digested tissue were isolated using the Satellite Cell Isolation Kit from Miltenyi Biotec, following the manufacturer's protocol.

## Transfection and immunostaining

A 96-well plate was coated in 10% Matrigel in DMEM for 1 min at RT. Matrigel was aspirated off and plates were incubated at 37°C and 5% $CO_2$ for 30 min, followed by 30 min at RT. Purified satellite cells were seeded at a density of 2000 cells/well. After 24 hr, cells were transfected with either the blank sgRNA plasmid, S100B sgRNA plasmid, or no plasmid using Lipofectamine 3000, according to the manufacturer's protocol. At 24 hr post transfection, transfected cells were fixed with 4% PFA in PBS for 20 min at RT. Blocking was performed as described above, and cells were incubated with primary antibody (1:200 dilution of anti-S100B) overnight at 4°C, followed by secondary antibody (1:500 dilution of AF647 anti-rabbit) for 2 hr at RT. Nuclei were stained with DAPI (1.5 µg/mL) in PBS for 1 min at RT. Immunolabeled cells were imaged on a Zeiss epifluorescent microscope using a 10× objective. Mean fluorescent intensity of S100B signal was calculated using MATLAB. Statistical comparison among cells transfected with no plasmid, the blank sgRNA plasmid, and the S100B sgRNA (n=4 culture wells, each with >100 cells) was performed using a one-way ANOVA (p = 2.08 × 10$^{-3}$). A post hoc Tukey test was performed (S100B sgRNA-blank sgRNA, p = 0.0634; S100B sgRNA-no plasmid control, p = 0.0016; blank sgRNA-no plasmid control, p = 0.078).

## TUNEL staining

Cell death in transfected cells was detected using In Situ Cell Death Detection Kit, TMR red, according to the manufacturer's protocol. Subsequently, nuclei were stained with DAPI (1.5 µg/mL) in PBS for 1 min at RT, and cells were imaged on a Zeiss epifluorescent microscope using a 10× objective. Mean fluorescent intensity of the TUNEL signal was calculated using MATLAB. Statistical comparison among cells transfected with no plasmid, the blank sgRNA plasmid, and the S100B sgRNA (n=4 culture wells, each with >100 cells) was performed using a one-way ANOVA (p = 2.32 × 10$^{-7}$). A post hoc Tukey test was performed (S100B sgRNA-blank sgRNA, p = 5.0 × 10$^{-4}$; S100B sgRNA-no plasmid control, p = 2.0 × 10$^{-7}$; blank sgRNA-no plasmid control, p = 2.0 × 10$^{-5}$).

## CellRox staining

Transfected cells were stained for ROS using CellROX Deep Red Reagent, according to the manufacturer's protocol. Nuclei were stained with Hoechst 33342 (5 µg/mL) in growth media for 30 min at 37°C and 5% $CO_2$. Cells were imaged on a Zeiss epifluorescent microscope using a 10× objective. Mean fluorescent intensity of the ROS signal was calculated using MATLAB. Statistical comparison

among cells transfected with no plasmid, the blank sgRNA plasmid, and the S100B sgRNA (n=5 culture wells, each with >100 cells) was performed using a one-way ANOVA (p = $2.33 \times 10^{-3}$). A post hoc Tukey test was performed (S100B sgRNA-blank sgRNA p = 0.0159; S100B sgRNA-no plasmid control, p = $2.3 \times 10^{-3}$; blank sgRNA-no plasmid control, p = 0.541).

## Myoblast differentiation

Twenty-four hours after transfecting cells with either the blank sgRNA plasmid, S100B sgRNA plasmid, or no plasmid, the transfection reagents and growth media were aspirated off and replaced with differentiation media (5% horse serum, 1% Penn/Strep in DMEM). Transfected cells were incubated in differentiation media for 3 days, then fixed with 4% PFA for 20 min at RT. Blocking was performed as described above, and cells were incubated with primary antibody (1:10 dilution of anti-MyHC) overnight at 4°C, followed by secondary antibody (1:200 AF647 goat anti-mouse IgG) for 2 hr at RT. Nuclei were stained with DAPI (1.5 µg/mL) in PBS for 1 min at RT. Immunolabeled cells were imaged on a Zeiss epifluorescent microscope using a 10× objective. Images were analyzed using ImageJ to determine the fusion index, calculated as number of nuclei in myoblasts divided by total number of nuclei. Statistical comparison of the fusion indexes among cells transfected with no plasmid, the blank sgRNA plasmid, and the S100B sgRNA (n=4 culture wells, each with >100 cells) was performed using a one-way ANOVA (p = $4.78 \times 10^{-4}$). A post hoc Tukey test was performed (S100B sgRNA-blank sgRNA, p = $3.0 \times 10^{-3}$; S100B sgRNA-no plasmid control, p = $5 \times 10^{-3}$; blank sgRNA-no plasmid control, p = 0.360).

## scRNA-Seq data processing and analysis

CellRanger v2.0 or v3.1 (10× Genomics) was used to process raw data. The CellRanger workflow aligns sequencing reads to the mm10 transcriptome using the STAR (*Dobin et al., 2013*) aligner and exports count data. The CellRanger count command was run with default parameters with the exception of the `–expect-cells` parameter which was set at 10,000. Filtered feature barcode data from all samples in HDF5 matrix format, both locally generated datasets and public 24-month TMS (*Tabula Muris Consortium and Almanzar, 2020*) datasets, were imported directly into LIGER (*Welch et al., 2019*) objects using the read10X import method. Only cells with a minimum of 300 genes expressed were retained during import. All cell counts provided in the main manuscript text are post-filtering counts. Seventeen datasets were generated in this manner: uninjured C57BL/6 WT female mice both young (3–4 months) and aged (20–24 months) datasets (seven datasets), and aged datasets at days 3 and 7 post injury (two datasets); Sod1$^{-/-}$ and SynTgSod1$^{-/-}$ rescue mouse datasets (3); public 24-month-old TMS skeletal muscle (2) datasets. Datasets were aggregated into the first two collections discussed in this manuscript using the createLiger function: the 'uninjured' collection (*Figure 3*) composed of uninjured young, aged, and TMS datasets; and the 'neurodegenerative' collection (*Figure 6*) using uninjured young, aged, Sod1 KO and rescue sets, and the TMS datasets. The merged LIGER objects were normalized, and variable genes were identified using selectGenes using default parameters before scaling with scaleNotCenter. Parameters k and lambda were selected using the suggest function and iterating through lambda parameters 1, 3, and 5 to maximize dataset alignment (calcAlignment) and agreement (calcAgreement). The first collection of datasets was batch-corrected using lambda = 3 and k = 15 (alignment = 0.96, agreement = 0.11), and the second collection was batch-corrected using lambda = 3, k = 30 (alignment = 0.98, agreement = 0.10). Other batch correction parameters were kept at their default values. Following UMAP dimensional reduction in LIGER, mitochondrial and ribosomal-dominated factors were removed from further analysis. LIGER objects were converted to Seurat (*Butler et al., 2018*) objects. Using Seurat, shared nearest neighbor (SNN) graphs were generated and cell clusters were identified based on the iNMF factors not dominated by mitochondrial or ribosomal genes. Clustering resolution was set at 1. All other parameters used were the defaults. Plots were generated in Seurat.

For several additional analysis discussed in the manuscript, subsets of cells were selected from each collection using the LIGER subset function, choosing cells with specific cluster identifications, followed by converting to Seurat objects and re-clustering of the subsets using the FindNeighbors and RunUMAP Seurat functions.

### Terminal Schwann cell vs. Pax7+ NMJ cell comparison

The Seurat object containing Pax7+NMJ cells was merged across columns to generate two pseudo-bulk RNA-Seq datasets per single cell RNA-Seq dataset. Thus, bulk dataset replicates were roughly based on the biological scRNA-Seq replicates (n=4 from two scRNA-Seq datasets, each containing cells from two mice). Five public bulk RNA-Seq datasets were aligned using Kallisto (*Bray et al., 2016*) and imported into R using tximport (*Soneson et al., 2016*). Transcript IDs were annotated with gene names using EnsDb.Mmusculus.v79, then samples were imported into DESeq2 (*Love et al., 2014*) for differential expression analysis with design = ~celltype. Genes were filtered based on expression in at least five cells, and differential expression analysis was otherwise performed using default parameters. Volcano plots were generated using the R package EnhancedVolcano.

### RNA velocity analysis

CellRanger v3.1 (10× Genomics) was used to process raw data. The CellRanger count command was run with default parameters with the exception of the —expect-cells parameter which was set at 10,000. Loom files were generated from the CellRanger 'outs' folders using the velocyto (*La Manno et al., 2018*) run10x function with the CellRanger mm10 v3.0.0 genome annotation file. Loom files were imported into Seurat objects with the ReadVelocity and as.Seurat functions from the SeuratWrapper package. Five datasets were generated this way, including two uninjured WT aged (22–26 months) datasets, one aged dataset from each of 3 and 7 dpi, and one public dataset from age-matched (24 months) myonuclei. Datasets were aggregated using the Seurat merge function, followed by normalization and identification of variable features separately by dataset. Batch effects were corrected for using CCA (canonical correlation analysis) with the FindIntegrationAnchors (dims = 1:10, k.filter = 50, anchor.features = 5000) followed by IntegrateData Seurat functions. Integrated data was scaled, and dimension reduction was performed using PCA and UMAP with the first 30 dimensions. SNN graphs and cell clusters were generated from the PCA-reduced data. The MuSC, myonuclei, and NMJ clusters were subset, followed by further subsetting of the NMJ cluster to obtain only NMJ cells expressing Pax7, and re-merging and integrating (anchor.features = 2000, dims = 1:20) the Pax7+ NMJ cells with the MuSCs and myonuclei. Data scaling and dimensional reduction (dims = 1:30), and clustering (resolution = 0.5) were repeated before exporting to an h5ad file with the SaveH5Seurat and Convert functions.

In Jupyter-notebook, the h5ad file was read into an scVelo (*Bergen et al., 2020*) AnnData object. Neighbors and moments were re-computed, followed by the velocities and velocity graph using default parameters. The velocity connection graph was generated with a threshold of 0.35.

## Acknowledgements

The authors thank Daniel Vincenz for assistance with artwork, the UM DNA Sequencing Core for assistance with single cell sequencing library preparation, Doug Millay for the single nuclei RNA sequencing datasets, Alex Shalek for critical advice regarding the single cell sequencing and data processing, Josh Welch for insights into bioinformatics analysis, Xia Jiang for technical assistance and members of the Aguilar, Brooks, and Jang laboratories. We also wish to thank Dr Arul M Chinnaiyan for providing SP with laboratory space. Funding research reported in this publication was partially supported by National Institute on Arthritis and Musculoskeletal Disease of the National Institutes of Health under award number P30 AR069620 (CAA, SVB) and R01 AR071379 (BDL), R61 AR078072 (BDL), the National Institute on Aging P01 AG051442 (SVB), the National Institute on Aging R01 AG051456 (JVC), the Breast Cancer Research Foundation (PJU, SDM), the 3M Foundation (CAA), American Federation for Aging Research Grant for Junior Faculty (CAA), the University of Michigan Geriatrics Center and National Institute of Aging under award number P30 AG024824 (CAA, SVB), the Department of Defense and Congressionally Directed Medical Research Program W81XWH2010336 (CAA, YCJ) and W81XWH-18-1-0653 and W81XWH-20-1-0795 (BDL), PCF Young Investigator Award and a Department of Defense Idea Development Award (SP), and the National Science Foundation Graduate Research Fellowship Program under grant number DGE 1256260 (JL). The content is solely the responsibility of the authors and does not necessarily represent the official views of the National Institutes of Health or National Science Foundation.

## Additional information

### Funding

| Funder | Grant reference number | Author |
|---|---|---|
| National Institute on Aging | P01 AG051442 | Susan V Brooks |
| National Institute on Aging | R01 AG051456 | Joe V Chakkalakal |
| National Institute of Arthritis and Musculoskeletal and Skin Diseases | P30 AR069620 | Susan V Brooks Carlos A Aguilar |
| National Institute of Arthritis and Musculoskeletal and Skin Diseases | R01 AR071379 | Benjamin Levi |
| 3M Foundation | | Carlos A Aguilar |
| American Federation for Aging Research | | Carlos A Aguilar |
| National Institute on Aging | P30 AG024824 | Susan V Brooks Carlos A Aguilar |
| Congressionally Directed Medical Research Programs | W81XWH2010336 | Young C Jang Carlos A Aguilar |
| Congressionally Directed Medical Research Programs | W81XWH-18-1-0653 OR170174 | Benjamin Levi |
| Breast Cancer Research Foundation | | Peter J Ulintz Sofia D Merajver |
| National Science Foundation | DGE 1256260 | Jacqueline A Larouche |
| Congressionally Directed Medical Research Programs | W81XWH-21-1-0492 | Benjamin Levi Carlos A Aguilar |

The funders had no role in study design, data collection and interpretation, or the decision to submit the work for publication.

### Author contributions

Jacqueline A Larouche, Conceptualization, Data curation, Software, Formal analysis, Investigation, Methodology, Writing - original draft, Writing - review and editing; Mahir Mohiuddin, Jeongmoon J Choi, Paula Fraczek, Sarah J Kurpiers, Jesus Castor-Macias, Wenxuan Liu, Robert Louis Hastings, Lemuel A Brown, James F Markworth, Kanishka De Silva, Investigation; Peter J Ulintz, Data curation, Software; Kaitlyn Sabin, Sethuramasundaram Pitchiaya, Investigation, Methodology; Benjamin Levi, Sofia D Merajver, Resources; Gregorio Valdez, Resources, Investigation; Joe V Chakkalakal, Resources, Investigation, Writing - review and editing; Young C Jang, Conceptualization, Resources, Formal analysis, Investigation, Methodology, Project administration, Writing - review and editing; Susan V Brooks, Conceptualization, Resources, Formal analysis, Supervision, Funding acquisition, Investigation, Methodology, Writing - original draft, Project administration, Writing - review and editing; Carlos A Aguilar, Conceptualization, Formal analysis, Supervision, Funding acquisition, Methodology, Writing - original draft, Project administration, Writing - review and editing

### Author ORCIDs

Jacqueline A Larouche (iD) https://orcid.org/0000-0001-9380-3547
Peter J Ulintz (iD) http://orcid.org/0000-0002-2037-8655
Gregorio Valdez (iD) http://orcid.org/0000-0002-0375-4532
Joe V Chakkalakal (iD) http://orcid.org/0000-0002-8440-7312
Young C Jang (iD) https://orcid.org/0000-0002-9489-2104
Carlos A Aguilar (iD) https://orcid.org/0000-0003-3830-0634

## Ethics

Animal experimentation: This study was performed in strict accordance with the recommendations in the Guide for the Care and Use of Laboratory Animals of the National Institutes of Health. All of the animals were handled according to approved institutional animal care and use committee (IACUC) protocols (IACUC protocol #: PRO00008428, PRO00006689) of the University of Michigan.

## Decision letter and Author response

Decision letter https://doi.org/10.7554/eLife.66749.sa1
Author response https://doi.org/10.7554/eLife.66749.sa2

---

# Additional files

## Supplementary files

• Transparent reporting form

## Data availability

Data have been deposited to GEO under accession code GSE165978.

The following dataset was generated:

| Author(s) | Year | Dataset title | Dataset URL | Database and Identifier |
|---|---|---|---|---|
| Larouche J, Mohiuddin M, Choi JJ, Ulintz PJ, Fraczek PM, Kurpiers SJ, Castor-Macias J, Liu W, Hastings RL, Brown LA, Markworth JF, Silva K, Levi BD, Merajver SD, Valdez G, Chakkalakal JV, Jang Y, Brooks S, Aguilar CA | 2021 | Muscle Stem Cell Response to Perturbations of the Neuromuscular Junction Are Attenuated With Aging | https://www.ncbi.nlm.nih.gov/geo/query/acc.cgi?acc=GSE165978 | NCBI Gene Expression Omnibus, GSE165978 |

The following previously published dataset was used:

| Author(s) | Year | Dataset title | Dataset URL | Database and Identifier |
|---|---|---|---|---|
| Petrany MJ, Swoboda CO, Sun C, Chetal K, Chen X, Weirauch MT, Salomonis N, Millay DP | 2020 | Single-nucleus RNA-seq identifies transcriptional heterogeneity in multinucleated skeletal myofibers | https://www.ncbi.nlm.nih.gov/geo/query/acc.cgi?acc=GSE147127 | NCBI Gene Expression Omnibus, GSE147127 |

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
