## [Decision Letter]

**Acceptance summary:**

This study provides important new insights into changes in the neuromuscular junction relevant to aging and disease. It contributes to understanding the dynamics at the neuromuscular junction and associated nearby stem cells, which could lead to new therapeutics as well as improve our basic understanding of aging and neuromuscular diseases. Overall this paper identifies a potentially new subtype of muscle stem cell with potential to improve our ability to modulate or study muscle stem cell changes in neurodegenerative diseases or aging.

**Decision letter after peer review:**

Thank you for submitting your article "Muscle Stem Cell Response to Perturbations of the Neuromuscular Junction Are Attenuated With Aging" for consideration by *eLife*. Your article has been reviewed by 3 peer reviewers, and the evaluation has been overseen by a Reviewing Editor and Matt Kaeberlein as the Senior Editor. The reviewers have opted to remain anonymous.

Summary:

The authors show that MuSCs near the NMJ adopt a modified phenotype during ageing and in Sod1 mutant model of neuromuscular degeneration. The main findings reported in this study are the following:

1) While total number of MuSCs was reduced in aged mice, NMJ-proximal MuSCs were comparable in young and aged mice. Synaptic myonuclei were also comparable in young and aged mice; 2) Muscle denervation induced by sciatic nerve transection (SNT) resulted in recruitment of MuSCs to the NMJ more efficiently in young than in aged mice; 3) scRNA-seq of MuSCs obtained from uninjured young and aged mice revealed expression of genes, such as Cadm1, Rapsn, and S100beta associated with neurite growth, NMJ maintenance, and repair in aged MuSCs; 4) Muscle injury induced a gene expression program distinct from that caused by muscle denervation; 5) scRNA-seq of MuSCs derived from superoxide dismutase Sod1-/- mice revealed commonalities between Sod1-/- and aged MuSCs gene programs; 6) Rescue experiments with synapsin driven Sod1 in motoneurons confirms the crosstalk between motoneuron, MuSC and synaptic myonuclei.

Essential revisions:

1) Although the study is interesting and the experiments well done, the manuscript is difficult to read. Adding a scheme that recapitulates the main results and the interactions between MuSC near the NMJ to illustrate the crosstalk between these different cell types would be helpful.

2) Supp Figure 1 and Figure 2. Aged mice have a 40% decrease in the overall number of MuSCs, compared to young mice. However, young and aged mice have a similar percentage of NMJ-proximal MuSCs, indicating a selective sparing of NMJ-proximal MuSC depletion in aged mice. Provided non-proximal NMJ MuSCs will migrate to the NMJ after SNT, it is expected that aged mice, having less non-proximal NMJ MuSCs that young mice, would accumulate less NMJ-proximal MuSCs. In Figure 2C, the percentage of MuSCs near NMJ is more in aged than in young. The authors should comment on these observations.

3) Figure 1D legend is mislabeled.

4) Figure 2- regarding the PAX7 cells near NMJs in young versus old stain with typical SC markers, are they quiescent or activated or myoblasts etc? How does this change +/- SNT?

5) Supp. Figure 2. Differentially expressed genes in the seven cell clusters should be reported.

6) Supp. Figure 2E Pax7+/S100b+ cells should be quantified.

7) Figure 3. What are the absolute expression levels for S100b, Rapsn, Chodl and other transcripts detected in aged non-MuSC and MuSC clusters? A minority of cells in the NMJ cluster expresses Pax7, Myf5, and Rapsn whereas S100b transcripts are detected in the vast majority of the cells (Figure 3D). Since the NMJ cluster derives from FACS-isolated MuSCs, what is the identity of S100b+/Pax7-/ cells?

8) The figure legend of Figure 3C is unclear – why are data from FACS in aged animals different from scRNA-seq aged data: it seems from the legend that the same cell population was compared either by IF or by RNA-seq. This should be better explained in the text. The differences observed are not clear; the sentence "largest non-MuSCs cluster of FACS sorted MuSCs "is confusing. Also, indicate clearly that the NMJ reclustering of Figure 3D was done with aged FSM (please change Sup 2F for 2E, and Sup 2E for 2D/rapsn).

9) Since the % of Pax7+S100B + observed in old mice is around 16%, and the % of SC near the NMJ in old mice 4%, does this mean that SCs outside the NMJ express S100B?

10) Do the Pax7+S100Beta+ SCs also express activated markers (MYF5, MYOD, MYOG) greater than Pax7+S100Beta- SCs ?

11) Stainings and qPCR with other SC markers in the Pax7+S100Beta subset should be included to confirm this subset is a true SC and not another lineage and validate sequencing data.

12) Conversely, are there other schwann cell markers expressed in the Pax7+S100Beta+ subset via IF or qPCR?

13) Figure 3E: Have cells from young animals been compared to cells from old animals? Please clarify.

14) Figure 3E. Is S100b present in the NMJ-associated stem cell transcripts?

15) In Figure 4B, what does the NMJ cluster in black represent: NMJ myonuclei as written? So, why does the RNA velocity not connect MuSCs with NMJ myonuclei?

16) In Figure 5d. The legend to the experiments described in Figure 5D reads: " D) UMAP dimensional reduction of FACS-isolated MuSCs (FSMs) from young and aged WT mice…..". The UMAP reported in Figure5d identifies clusters assigned to MuSCs, Immune, Endo, MSC, SMC, and other cells types. Please explain how cell lineages (immune, smooth muscle, endothelial cells) other than MuSCs were FACS-purified using a scheme designed to selectively purify MuSCs? There is no overlap or contiguity between MuSCs and NMJ (and other cell types) indicating sharply distinct transcriptomes of different clusters.

17) Is Sod1 only expressed in motoneurons or also in their associated Schwann cells in the SynTgSod1model?

18) A better integration of the data from the two models, aging and Sod 1 -/-, should be included to decipher similarities/differences in these two models with regard to SC behavior.

19) What portion of the aged mice have similar defects in MNs or at the NMJ? In other words, how similar are the models?

20) The authors should comment on the mechanism/s by which MuSCs turn on NMJ transcripts. Are they acquired over time or immediately upon injury? Do they stay on or change over time?

21) The small letters of the names of the genes in Figure 5F are difficult to read.

22) Do the authors think that MuSCs near the NMJ stay in their initial location without moving along the myofiber in both young and old animals?

23) Is the Pax7+S100B+ phenotype of MuSC a transient state, or could it represent a signature of specialized MuSC?

24) Expansion ex vivo of Pax7+S100B-GFP+ MuSC should be performed to test the possibility that these myogenic stem cells have acquired the specific identity of NMJ MuSC.

25) In the genetic rescue of motor neurons in Sod1-/- mice, what is percentage of MN rescue seen when MN are provided, and is there a threshold effect?

26) If Pax7+S100Beta+ cells are lost or blocked in either model, does this alter rescue effects? In other words, is it responsible in part for phenotypes in models or bystander?

27) It has been proposed that a specific class of fibroblasts near the tips of the myofiber can acquire a myogenic identity and contribute to muscle growth (https://www.biorxiv.org/content/10.1101/2020.07.20.211342v1, https://www.biorxiv.org/content/10.1101/2020.07.20.213199v1). Can the authors exclude the possibility that Schwann cells that share many common expressed genes with NMJ MuSC could be reprogrammed to adopt a myogenic fate at the NMJ, or vice versa?

28) Kimmel et al. (Development 2020) have reported that the transcriptomes of young and old MuSCs are hardly distinguishable. How do the MuSCs transcriptomes reported in this study compare with those reported by Kimmel et al.?

[Editors' note: further revisions were suggested prior to acceptance, as described below.]

Thank you for resubmitting your work entitled "Murine Muscle Stem Cell Response to Perturbations of the Neuromuscular Junction Are Attenuated With Aging" for further consideration by *eLife*. Your revised article has been evaluated by Matt Kaeberlein as the Senior Editor, a Reviewing Editor, and two of the original reviewers.

The manuscript has been improved but there are some remaining issues that need to be addressed, as outlined below:

1) Please specify whether the myofiber atrophy observed after SNT is linked to definitive muscle denervation, or if the SNT allows the re-innervation of the muscles during the 28 days following injury. In the first case, the accretion of SC near the NMJ may act to counteract the expected muscle atrophy following denervation (leading to denervated myofibers with an increased number of myonuclei), while in the second case the presence of newly accreted myonuclei near the NMJ could reflect a late re-innervation process.

The authors should explain this when presenting the SNT model used to alleviate ambiguities. Furthermore the re-innervation process is more impaired in old and Sod1-/- animals, which could lead to the decreased SyM observed. If this is the case, SNT should lead to a more severe muscle atrophy in old animals than in young animals; this should be referenced and discussed.

Concerning the Figure 1C: the authors should establish the number of myonuclei in the myofiber 28 days after nerve injury: from their picture there is a massive fusion of SC (Pax7-nGFP); It is quite unexpected that during atrophy induced by nerve section, there is such an increase of new myonuclei: please indicate their number.

The authors need to discuss nerve regrowth in their 28 days SNT paradigm Specifically, it important to indicate that this increase of new synaptic myonuclei is due to nerve regrowth.

2) In figure 2D, why is the myofiber around the NMJ after SNT not Tomato+ (accretion of SC expressing TdTomato), while Synaptic myonuclei (SyM) are GFP+ in Figure 1C? The authors write (line 135) that tamoxifen was administered after SNT, but no information concerning when tamoxifen was administered is provided: 1 day after SNT? 25 days after SNT? Please clarify.

3) Probes used to detect PolyA RNA (Figure 3E) are not indicated.

4) In Figure 3D, could the authors add a three color (Young/Tabula Muris/Aged) re-clustered UMAP diagram, to better identify Tabula Muris/young/aged Nrn1+ and S100b + cells?

5) Legend of Figure 3 Sup 2A: "…not enough Pax7+S100B+ (young?) cells were captured…was young was omitted in the legend?

---

## [Author Response]

Essential revisions:1) Although the study is interesting and the experiments well done, the manuscript is difficult to read. Adding a scheme that recapitulates the main results and the interactions between MuSC near the NMJ to illustrate the crosstalk between these different cell types would be helpful.

We thank the reviewer for bringing this to our attention. We have clarified text in the manuscript that was difficult to read and added a new Figure (Figure 7) that integrates our findings and proposes a model for the observed behavior.

2) Supp Figure 1 and Figure 2. Aged mice have a 40% decrease in the overall number of MuSCs, compared to young mice. However, young and aged mice have a similar percentage of NMJ-proximal MuSCs, indicating a selective sparing of NMJ-proximal MuSC depletion in aged mice. Provided non-proximal NMJ MuSCs will migrate to the NMJ after SNT, it is expected that aged mice, having less non-proximal NMJ MuSCs that young mice, would accumulate less NMJ-proximal MuSCs. In Figure 2C, the percentage of MuSCs near NMJ is more in aged than in young. The authors should comment on these observations.

We thank the reviewer for this excellent comment. We agree that since aged mice have less MuSCs overall, we anticipated less would accumulate near the NMJ after nerve injury when compared to young MuSCs. However, we found the opposite behavior and attribute this variation to changes in the ability of young MuSCs to engraft into synaptic myonuclei compared to aged MuSCs. We enumerated synaptic myonuclei for both young and aged muscles and observed a statistically significant increase in the number of synaptic myonuclei for young muscle compared to aged. We elaborate further in the discussion on possible mechanisms that inhibit aged MuSCs from differentiating compared to young MuSCs.

3) Figure 1D legend is mislabeled.

This has been amended and the correct label added.

4) Figure 2 – regarding the PAX7 cells near NMJs in young versus old stain with typical SC markers, are they quiescent or activated or myoblasts etc? How does this change +/- SNT?

We thank the reviewer for this insightful question. In our Pax7-tdTomato model, we administered tamoxifen chow 2 weeks following the SNT injury, which is after most MuSCs typically undergo activation and differentiation following a denervation injury (Borisov et al., 2005 and reference 44). This allowed us to visualize and quantify MuSCs expressing Pax7^+^ 28 days following SNT, rather than MuSC progeny that had already undergone myogenesis. We performed immunostaining for MyoD on myofibers after SNT and did not observe appreciable expression above background. This result indicates that these MuSCs are quiescent satellite cells. In further support of these cells being quiescent, several studies have established that exposure to neurotrophic factors such as GDNF (Li et al. 2019 and reference 62) and BDNF (Mousavi et al. 2006 and reference 58) inhibits MuSC differentiation. This is discussed further in the Discussion section.

5) Supp. Figure 2. Differentially expressed genes in the seven cell clusters should be reported.

We have modified Figure 2 – Supp. Figure 1 and included a heatmap (new Figure 2 – Supp. Figure 1d) describing the top 5 uniquely expressed genes in the different cell clusters.

6) Supp. Figure 2E Pax7+/S100b+ cells should be quantified.

We apologize for the lack of clarity. The quantifications for these images are in Figure 3c, and we have amended the figure captions of Figures 3c, 5e, and Figure 2 – Supp. Figure 2b and Figure 5 – Supp. Figure 2a to reflect this.

7) Figure 3. What are the absolute expression levels for S100b, Rapsn, Chodl and other transcripts detected in aged non-MuSC and MuSC clusters? A minority of cells in the NMJ cluster expresses Pax7, Myf5, and Rapsn whereas S100b transcripts are detected in the vast majority of the cells (Figure 3D). Since the NMJ cluster derives from FACS-isolated MuSCs, what is the identity of S100b+/Pax7-/ cells?

This is an excellent point brought up by the reviewer. To further validate the absolute expression levels of several synaptic genes in MuSCs, we have performed single molecule RNA fluorescent in situ hybridization (smRNA-FISH) on FACS isolated MuSCs from young and aged muscle and observed excellent agreement with our sequencing data. This data is shown in a new Figure 3F and Figure 3 – Supp. Figure 2D. We have also modified Figure 3 – Supp. Figure 2 and added a new supplemental figure (Figure 3 – Supp. Figure 2C) that further details the expression of synaptic-associated genes in the NMJ and MuSC clusters for both MuSCs and non-MuSCs in the NMJ cluster (defined by cells that express Pax7 as MuSCs and cells that do not express Pax7 as non-MuSCs). We comment in the manuscript that we believe other cells in the NMJ cluster are most likely Schwann cells and/or glia.

8) The figure legend of Figure 3C is unclear – why are data from FACS in aged animals different from scRNA-seq aged data: it seems from the legend that the same cell population was compared either by IF or by RNA-seq. This should be better explained in the text. The differences observed are not clear; the sentence "largest non-MuSCs cluster of FACS sorted MuSCs "is confusing. Also, indicate clearly that the NMJ reclustering of Figure 3D was done with aged FSM (please change Sup 2F for 2E, and Sup 2E for 2D/rapsn).

We are very appreciative of the reviewer for bringing this to our attention. The stacked bar graphs in Figure 3c are comparing scRNA-Seq quantification and IF quantification of Pax7+ and Pax7+S100b+ cells among FACS enriched MuSCs from young and aged tissues. We have modified the x-axis labels on the figure as well as the descriptions of the results in the text. Additionally, we have re-worded the text to clarify that the majority of FSMs that did not cluster with the MuSC cluster instead clustered with cells expressing NMJ transcripts. Differences between IF and scRNA-Seq quantifications could be the result of differences in mRNA vs protein expression, and/or transcript abundance. Lastly, NMJ re-clustering was done with the entire NMJ cluster, and thus has both aged and young FACS enriched MuSCs. Since more aged FACS enriched MuSCs clustered with the original NMJ cluster, they are over-represented in the NMJ re-clustering compared to young FACS enriched MuSCs. We have clarified this in the text.

9) Since the % of Pax7+S100B + observed in old mice is around 16%, and the % of SC near the NMJ in old mice 4%, does this mean that SCs outside the NMJ express S100B?

We thank the reviewer for bringing this to our attention. In our initial quantification of 4% MuSCs near the NMJ, we used stringent criteria to define “near” the NMJ as within 50 µm. However, we have quantified the data again using more representative criteria of MuSCs that can interact with the NMJ by considering MuSCs within 250 µm as near the NMJ or within the middle region of the myofiber. With this quantification, we observed approximately 18% of MuSCs in aged mice as near the NMJ. Furthermore, we performed immuno-staining for Pax7 on S100B-GFP myofibers isolated from young mice (6 months), and observed that the majority of S100B-expressing cells and all of the S100B^+^/Pax7^+^ cells are localized near the NMJ prior to SNT. Following SNT, we observed most S100B+ and S100B^+^/Pax7^+^ cells were still near the NMJ, but also a fraction of cells along the length of the myofiber, which is consistent with the known role of Schwann cells to guide regenerating axons.

10) Do the Pax7+S100Beta+ SCs also express activated markers (MYF5, MYOD, MYOG) greater than Pax7+S100Beta- SCs ?

We thank the reviewer for this excellent point. In the new Figure 3 – Supp. Figure 2A, we have quantified the expression level of myogenic markers for the Pax7+S100Beta+ MuSCs and Pax7+S100Beta- MuSCs and they are approximately comparable in expression level. We note however that these MuSCs were FACS isolated and the isolation of MuSCs by FACS will induce some level of activation (Machado et al. Cell Reports 2017 and van Velthoven et al., Cell Reports 2017).

11) Stainings and qPCR with other SC markers in the Pax7+S100Beta subset should be included to confirm this subset is a true SC and not another lineage and validate sequencing data.

We are very appreciative for this comment from the reviewer. To further validate a subset of aged MuSCs express synaptic transcripts, we performed smRNA-FISH on FACS isolated MuSCs. We observed expression of Pax7 at a similar level to other smFISH experiments (Kann, A., et al. Develop. 146, dev179259, 2020). Next, we quantified a subset of synaptic transcripts in young and aged MuSCs and observed SynCam (Cadm1), Rapsyn (Rapsn), Neuritin 1 (Nrn1) and Cholinergic Receptor Nicotinic Alpha 1 Subunit (Chrna1) were expressed at approximately comparable levels to those observed with our sequencing data (R=0.88). These cells also expressed MyoD1, indicating these cells are myogenic, and strongly expressed Fos, which is associated with activated MuSCs. Thus, this subset of cells express both markers of myogenic progenitors and synaptic transcripts. However, as noted above these MuSCs were FACS isolated, which induces activation. These data are added into the new Figure 3 – Supp. Figure 2D, and Figure 3E-F.

12) Conversely, are there other schwann cell markers expressed in the Pax7+S100Beta+ subset via IF or qPCR?

We thank the reviewer for this comment. We have performed smRNA-FISH on FACS isolated MuSCs and validated several markers nominally associated with Schwann cells (SynCam and Neuritin 1) are expressed in Pax7+ MuSCs. These data are added into the new Figure 3E-F and Figure 3 – Supp. Figure 2D.

13) Figure 3E: Have cells from young animals been compared to cells from old animals? Please clarify.

This is an excellent point raised by the reviewer. We did not recover enough Pax7+S100b+ cells from young mice to do an appropriate comparison to aged Pax7+S100b+ cells. Our comparative analysis between Pax7+/S100b+ MuSCs and Pax7-/S100b+ cells was performed using all Pax7+ NMJ cells regardless of animal age, though most cells are from aged animals.

14) Figure 3E. Is S100b present in the NMJ-associated stem cell transcripts?

We thank the reviewer for this comment. We observed that S100B is present in both the Pax7+ cells as well as the Pax7- cells in the NMJ cluster in Figure 3E, albeit S100B is expressed at a lower level in Pax7+ cells. We have added additional plots in Figure 3 – Supp. Figure 2C whereby NMJ cells were split based on Pax7 expression, and the expression of MuSC specific genes, co-expressed genes, including S100b, and Schwann cell specific genes was compared. We have also added two lines in the results on these observations.

15) In Figure 4B, what does the NMJ cluster in black represent: NMJ myonuclei as written? So, why does the RNA velocity not connect MuSCs with NMJ myonuclei?

We thank the reviewer for this comment. The NMJ cluster in black in Figure 4B (now Figure 5B) corresponds to cells that express NMJ-associated genes and Pax7 (Pax7+S100B+). The yellow cells in that cluster also express both NMJ-associated genes, but were captured in the snRNA-Seq dataset rather than the scRNA-Seq datasets. We have amended the cluster labels to further clarify this.

The RNA velocity trajectories do not connect the main MuSC cluster to the NMJ-Pax7+ cluster because of differences in the transcriptional landscape and splicing kinetics compared to the main MuSC and myonuclei clusters. Velocity-inferred cell connections are shown in Figure 4 – Supp. Figure 1 and do connect the MuSC cluster to the NMJ cells, though the connections are sparse compared to cell-cell connections within the MuSC cluster, reflecting this discrepancy in transcriptional profiles and splicing.

16) In Figure 5d. The legend to the experiments described in Figure 5D reads: " D) UMAP dimensional reduction of FACS-isolated MuSCs (FSMs) from young and aged WT mice…..". The UMAP reported in Figure 5d identifies clusters assigned to MuSCs, Immune, Endo, MSC, SMC, and other cells types. Please explain how cell lineages (immune, smooth muscle, endothelial cells) other than MuSCs were FACS-purified using a scheme designed to selectively purify MuSCs? There is no overlap or contiguity between MuSCs and NMJ (and other cell types) indicating sharply distinct transcriptomes of different clusters.

This is a good point brought up by the reviewer and we have clarified this point in the text. Briefly, we used the Tabula Muris Senis limb muscle datasets from 24-month-old mice, which contained all mononucleated cells, to identify cells that were enriched using FACS and underwent scRNA-Seq but did not cluster with MuSCs. We posited that cells that were enriched with FACS and that do not cluster with the MuSCs were likely impurities that resulted from the FACS enrichment. To confirm our FACS enrichment of cells was not enriching other cell lineages, we immuno-stained and imaged FACS enriched cells for Pax7 in injured muscle. We observed ~95% of cells from the young muscle and ~90% of cells from the aged muscle express Pax7, which is consistent with previous reports that these markers isolate MuSCs (Maesner et al. Skeletal Muscle 6, 2016). These results are quantified in Figure 3C, and are consistent with the fraction of FACS enriched MuSCs that cluster with MuSCs profiled with scRNA-Seq. After 3 days post muscle injury with BaCl_2_ injection, we observed 90.6% of FACS enriched cells and underwent scRNA-Seq cluster with MuSCs. After 7 days post muscle injury with BaCl_2_ injection, we observed 94.6% of FACS enriched cells and underwent scRNA-Seq cluster with MuSCs.

17) Is Sod1 only expressed in motoneurons or also in their associated Schwann cells in the SynTgSod1model?

We thank the reviewer for this comment. Synapsin is a neuron-specific protein associated with synaptic vesicles, and human Sod1 (hSod1) is expressed in all neurons including motoneurons in the rescue model. However, hSod1 is not expressed in Schwann cells.

18) A better integration of the data from the two models, aging and Sod 1 -/-, should be included to decipher similarities/differences in these two models with regard to SC behavior.

We thank the reviewer for this suggestion. We performed an additional differential gene expression analysis comparing the MuSC scRNA-Seq cluster across datasets (young, aged, knockout and rescue) and have included it in Figure 5 – Supp. Figure 1e. Clustering analysis (BuildClusterTree) of the different models (Young, Aged, Sod1-/-, and SynTgSod1-/-) using all highly variable genes showed Aged and Sod1-/- MuSCs are transcriptionally the most similar, followed by SynTgSod1-/-, and Young MuSCs are the most transcriptionally distinct (Author response image 1). This is in line with the partial rejuvenation that we observed in SynTgSod1-/- NMJs compared to Sod1-/- NMJs. Both aged and Sod1-/- MuSCs displayed enrichments in stress response genes (Gadd45a, Jund) associated with muscle atrophy, as to where young MuSCs displayed more of an activated expression profile (Myod1, Fosl1). The SynTgSod1-/- MuSCs were enriched for genes associated with ribosomal activity and protein translation (Rack1, Nop53, Eef1g).

**Author response image 1. sa2fig1:** Dendrogram of transcriptional similarity across models. SOD1.KO: SOD1-/-, SOD1.Rescue: SynTgSod1-/-.

19) What portion of the aged mice have similar defects in MNs or at the NMJ? In other words, how similar are the models?

We thank the reviewer for this question. In our revised Figure 2 – Supp. Figure 1, we show quantification of NMJ fragmentation and structure in young and aged mice. We observed young and aged mice contained approximately similar numbers of post-synaptic myonuclei, and synaptic area and volume. However, aged NMJs were observed with increased fragmentation of acetyl choline receptors (AChRs, new Figure 2 – Supp. Figure 1e). These results indicate some similarity between the two models but increased NMJ degeneration in aged muscles.

20) The authors should comment on the mechanism/s by which MuSCs turn on NMJ transcripts. Are they acquired over time or immediately upon injury? Do they stay on or change over time?

We thank the reviewer for this comment. In our revised manuscript, we have performed additional analysis testing the hypothesis that MuSCs increase expression of NMJ-associated transcripts through chronic exposure to signaling that is normally restricted to the synapse such as from Schwann cells. We used a young (6 months) murine model that contains a fluorescent reporter for S100β (S100β–GFP, Figure 6a) and harvested uninjured myofibers/myobundles as well as 28 days after sciatic nerve transection from TA muscles (Figure 6b). We observed an increase in the number of *S100β^+^* cells outside of nerve terminals after SNT, which was consistent with the known role of Schwann cells to guide regenerating axons. We also observed an increase in the number of Pax7+/S100β+ cells and these were co-localized with Pax7-/S100β+ cells indicating enhanced interactions between MuSCs and Schwann cells occur as a result of denervation. We speculate that the increase in expression of synaptic transcripts in MuSCs may be the result of increased exposure to neurotrophins. In further support of this, our datasets which are consistent with others (Hicks et al. Nature Cell Biol. 2018, Mousavi et al. J. Neurosci. 2006), show the subset of MuSCs that express synaptic transcripts also showed increased expression of neurotrophic receptors such as ERBB3 and NGFr (Figure 6 – Supp. Figure 1D). We hypothesize that the development of this fate does not occur immediately but rather in a gradual manner. This is due to our scRNA-Seq datasets after muscle injury, which do not show acquisition of a synaptic fate after 7 days. We comment further on these results in the discussion.

21) The small letters of the names of the genes in Figure 5F are difficult to read.

We thank the reviewers for pointing this out. We have increased the text size for this figure.

22) Do the authors think that MuSCs near the NMJ stay in their initial location without moving along the myofiber in both young and old animals?

This is a good point brought up by the reviewer. Previous research on MuSCs in vivo with intravital imaging (Webster et al. Cell Stem Cell 2016) has shown that MuSCs are immobile prior to injury. We do not have experimental results to show in uninjured muscle that MuSCs migrate towards the NMJ niche or are restrained from movement along a myofiber near the NMJ. We speculate that MuSCs would be drawn to the NMJ niche from trophic factors produced locally by Schwann cells, glia, and myonuclei such as glial cell line-derived neurotrophic factor (GDNF), brain-derived neurotrophic factor (BDNF), neurotrophin-3 (NT3), nerve growth factor (NGF), and vascular endothelial growth factor (VEGF). We have not validated this hypothesis yet, and will test this further in future work.

23) Is the Pax7+S100B+ phenotype of MuSC a transient state, or could it represent a signature of specialized MuSC?

We thank for the reviewer for this comment. We did not observe appreciable expression of S100B in the primary MuSC cluster generated through scRNA-Seq before or after muscle injury (Figure 3 – Supp. Figure 2a) and only observed S100B expression in NMJ associated cells. Thus, we hypothesize the population of MuSCs that are Pax7+S100B+ may be specialized for a synaptic fate. We performed a subset of experiments on MuSCs that increase expression of S100B in vitro (described further below); however, the hypothesis that S100B+ MuSCs take a synaptic fate has not been validated and we believe that such an experiment should be performed as part of a larger study designed to assess whether intrinsic or extrinsic factors primarily drive Pax7+S100B+ MuSC identity. For instance, it is possible that Pax7+S100B+ MuSCs may not retain nor acquire an NMJ identity without co-culturing with motor neurons, muscle fibers, Schwann cells and/or cell types together in addition to changing in response to neurodegeneration and/or aging.

24) Expansion ex vivo of Pax7+S100B-GFP+ MuSC should be performed to test the possibility that these myogenic stem cells have acquired the specific identity of NMJ MuSC.

We thank the reviewer for this comment. We do agree with the reviewer that culture experiments will provide additional insights about the identity of Pax7+S100B-GFP+ MuSCs. Given this subset of MuSCs was rare, we isolated primary MuSCs from young mice and activated S100B using a CRISPR-dCas9-SUNTAG transcriptional activator using short guide RNAs targeting the S100B locus (new Figure 6D-G). We first observed that increased activation of S100B induces apoptosis, which is consistent with previous observations (Sorci et al., J. Cell Physiol. 2004). Moreover, we detected increases in reactive oxygen species (ROS) and reductions in fusion as a result of S100B overexpression. Given the longer time frames through which MuSCs were proposed to adopt a synaptic fate, it was difficult to expand these cells without significant cell death.

25) In the genetic rescue of motor neurons in Sod1-/- mice, what is percentage of MN rescue seen when MN are provided, and is there a threshold effect?

We appreciate this comment from the reviewer. Recently we published that overexpression of human Sod1 (hSod1) in neurons demonstrates a nearly complete rescue of innervation that is lost with the Sod1-/- model for age-matched animals (Su et al., Free Rad. Biol. And Med. 2021). We have not measured a threshold effect but recovery of hSod1 to ~20-25% of WT protein levels in motoneurons provides rescue. We have included this reference in our revised Results section.

26) If Pax7+S100Beta+ cells are lost or blocked in either model, does this alter rescue effects? In other words, is it responsible in part for phenotypes in models or bystander?

This is an excellent point brought up by the reviewer. The rescue in the Sod1-/- model occurs only in motor neurons and is independent of MuSCs.

27) It has been proposed that a specific class of fibroblasts near the tips of the myofiber can acquire a myogenic identity and contribute to muscle growth (https://www.biorxiv.org/content/10.1101/2020.07.20.211342v1, https://www.biorxiv.org/content/10.1101/2020.07.20.213199v1). Can the authors exclude the possibility that Schwann cells that share many common expressed genes with NMJ MuSC could be reprogrammed to adopt a myogenic fate at the NMJ, or vice versa?

This is an interesting point brought up about the reviewer. To our knowledge, Schwann cells have not been observed to express myogenic transcription factors such as Pax7, MyoD1 and Myf5 or postsynaptic proteins such as Rapsyn or AChRs, and our datasets show Pax7-S100B+ cells do not express these factors (Figure 3 – Supp. Figure 2C). Additionally, we do not observe an appreciable fraction of Pax7+S100B+ cells in young muscle, indicating de-differentiation of Schwann cells into an alternate lineage would only occur in old age or diseased muscle. Given aged Schwann cells have been shown to have impaired de-differentiation ability (Painter et al., Neuron 2014), we posit that reprogramming of Schwann cells into myogenic progenitors is unlikely. Conversely, muscle stem cells have been shown to express different neurotrophic factors in a similar manner as Schwann cells and have been shown to differentiate into alternate lineages such as brown adipocytes through increased expression of the methyltransferase Prdm16 (Seale et al., Nature 2008). Additionally, S100B overexpression has been shown to promote a myoblast–brown adipocyte transition (Morozzi et al. Cell Death Differ. 2017).

28) Kimmel et al. (Development 2020) have reported that the transcriptomes of young and old MuSCs are hardly distinguishable. How do the MuSCs transcriptomes reported in this study compare with those reported by Kimmel et al.?

We thank the reviewers for this suggestion and have retrieved and integrated the datasets from Kimmel et al. with ours using integrative nonnegative matrix factorization. We observed substantial overlap indicating strong transcriptional similarity among the MuSC transcriptomes (Author response image 2). We did not observe many (n=4) Pax7+S100b+ single cell libraries (Author response image 2) in the Kimmel et al. manuscript but this could be the result of sorting on different surface markers (Integrin α7 and VCAM1 compared to CXCR4 and Integrin β1), or it may be related to lower counts and features recovered with the 10x v2 chemistry, as the majority of Pax7+S100b+ cells were among the activated cells, which have higher feature and count numbers more similar to the v3 libraries reported herein, from which we observed the Pax7+S100b+ cells. We have included several plots demonstrating this in Author response image 2.

**Author response image 2. sa2fig2:** (A) UMAP overlays of transcriptomes reported in Kimmel et al., Development 2020 (peach) and the datasets reported in this manuscript. (B) Marker-gene overlays showing canonical myogenic regulatory factors as well as S100b showing S100b is confined to the UMAP cluster containing primarily cells reported in this manuscript. (C) Pearson correlation coefficients across datasets reported herein as well as in Kimmel et al. shows high similarity overall, with three groups of highest correlation being the activated cells from Kimmel et al., our aged v3 datasets, and the young and aged quiescent v2 datasets. Some of this grouping is likely a result of differences in features and counts among the libraries. (D) Stacked bar graph showing origin of Pax7+S100b+ cells. Most are recovered in the aged v3 datasets we report, though several are also among the datasets reported by Kimmel et al., and in our young v3 dataset.

[Editors' note: further revisions were suggested prior to acceptance, as described below.]

The manuscript has been improved but there are some remaining issues that need to be addressed, as outlined below:1) Please specify whether the myofiber atrophy observed after SNT is linked to definitive muscle denervation, or if the SNT allows the re-innervation of the muscles during the 28 days following injury. In the first case, the accretion of SC near the NMJ may act to counteract the expected muscle atrophy following denervation (leading to denervated myofibers with an increased number of myonuclei), while in the second case the presence of newly accreted myonuclei near the NMJ could reflect a late re-innervation process.

We thank for the reviewer for this comment. Previously, we used the same SNT procedure to disrupt lower-limb NMJs (Liu et al., *eLife* 2015). We observed this injury results in complete denervation of adult NMJs, and re-innervation occurs 4–6 weeks after SNT as measured by immunofluorescence (IF) and physiological measures. Other published reports (Williams and Valdez et al., Science 2009) that used the same SNT procedure observed a significant fraction of mouse adult NMJs are re-innervated 3-6 weeks after SNT, which is approximately consistent with our previous findings (Liu et al., *eLife* 2015). Thus, we believe 4 weeks after SNT is a re-innervation phase whereby axon terminals return to myofibers and function is beginning to be restored.

Myonuclear accretion in the context of SNT could manifest as an increase in myonuclear number, turnover of myonuclei, or some combination of the two. Previously, we found a modest reduction of myonuclei 6 weeks after SNT that was exacerbated with MuSC depletion (supplemental data Liu et al., *eLife* 2015). This result was consistent with previous work no loss of myonuclei in chronically denervated muscle (Bruusgaard et al., J. Clin. Investig. 2008). We concluded that this supported loss and some level of myonuclear turnover after SNT, and these data were published and discussed (Liu et al., *eLife* 2015). We have included more details in the discussion in this regard.

To glean if engraftment of MuSCs occurs acutely after denervation or in response to a late re-innervation process, we performed lineage-tracing of MuSCs at different points after SNT. In Figure 1, we administered tamoxifen before SNT to facilitate tracing of the acute response of MuSCs to SNT. In Figure 2, we administered tamoxifen 14 days after SNT to be able to perform tracing of MuSCs in the late phases of re-innervation and discern MuSCs from the underlying fiber. We did not observe significant fusion of MuSCs >14 days post SNT and myofibers were devoid of TdTomato+ signal. It is possible the fusion of MuSCs into NMJ proximal locations are attempting to prevent atrophy and we discuss this and other potential roles the cells may perform in the discussion.

The authors should explain this when presenting the SNT model used to alleviate ambiguities. Furthermore the re-innervation process is more impaired in old and Sod1-/- animals, which could lead to the decreased SyM observed. If this is the case, SNT should lead to a more severe muscle atrophy in old animals than in young animals; this should be referenced and discussed.

We appreciate this comment from the reviewer. We agree that aged animals have been shown to display impairments in re-innervation when compared to young animals and this may contribute to the reductions in SyM we have observed. We do also note that this effect may be driven by several factors beyond aged muscle, MuSCs and nerve such as diminished Schwann cell responses (Painter et al., Neuron 2014). Previously, we found that MuSC depletion led to a reduction in SyM number, but not complete loss of SyM 6 weeks after SNT. This loss of MuSC contribution to SyM was also associated with reduced re-innervation of adult NMJs after SNT (Liu et al., *eLife* 2015). These studies coupled with our data showing fusion near the NMJ, as well as that a specific subset of MuSCs that express synaptic factors but do not engraft in old age and neurodegeneration suggest these cells are critical for re-innervation of muscle. We have included several additional references and sentences in the discussion about this.

Concerning the Figure 1C: the authors should establish the number of myonuclei in the myofiber 28 days after nerve injury: from their picture there is a massive fusion of SC (Pax7-nGFP); It is quite unexpected that during atrophy induced by nerve section, there is such an increase of new myonuclei: please indicate their number.

Please see the comments above regarding myonucler accretion. In regards to the fusion of MuSCs, we state in the manuscript that existing synaptic myonuclei and extra-synaptic myonuclei may incorporate nGFP after MuSC engraftment. We also mention that diffusion of molecules near the NMJ is limited to prevent synaptic transcription in myonuclei outside of the junctional area.

The authors need to discuss nerve regrowth in their 28 days SNT paradigm Specifically, it important to indicate that this increase of new synaptic myonuclei is due to nerve regrowth.

As noted above, we (Liu et al., *eLife* 2015) and others (Williams and Valdez et al., Science 2009) have previously observed that most adult NMJs are re-innervated 4-6 weeks after SNT. We have added several sentences to the results and Discussion section expanding on this.

2) In figure 2D, why is the myofiber around the NMJ after SNT not Tomato+ (accretion of SC expressing TdTomato), while Synaptic myonuclei (SyM) are GFP+ in Figure 1C? The authors write (line 135) that tamoxifen was administered after SNT, but no information concerning when tamoxifen was administered is provided: 1 day after SNT? 25 days after SNT? Please clarify.

We apologize for the lack of clarity in our presentation. The mouse models in Figures 1 and 2 are derived from different lineage tracing systems (Pax7^CreER^-Rosa26^nTnG^ for Figure 1 and Pax7^CreER^-Rosa26^TdTomato^ for Figure 2). In Figure 2, we utilized young and aged mice that harbor a MuSC fluorescent reporter that permits lineage tracing of the entire cell rather than just the nucleus (Pax7^CreER/+^-Rosa26^TdTomato/+^), which was critical to quantify the distances between MuSCs and the NMJ. The mice were administered tamoxifen at 14 days post sciatic nerve transection (SNT). We have modified the text in the Results section and the supplemental information on the animal model to reflect this. We quantified the recombination efficiency of TdTomato in Pax7+ cells to be >90% (data not shown). The myofiber around the NMJ is not TdTomato+ suggests engrafted nuclei from MuSCs occur earlier than 14 days.

3) Probes used to detect PolyA RNA (Figure 3E) are not indicated.

We have included the oligos in the table for the smRNA-FISH probes. Briefly, the polyA RNA probe is an Oligo-dT30 coupled to Alexa Fluor 488 (Oligo-dT30-AF488, 5` to 3`).

4) In Figure 3D, could the authors add a three color (Young/Tabula Muris/Aged) re-clustered UMAP diagram, to better identify Tabula Muris/young/aged Nrn1+ and S100b + cells?

We have added the 3 color plot of the re-clustered NMJ UMAP in a panel to Figure 3B.

5) Legend of Figure 3 Sup 2A: "…not enough Pax7+S100B+ (young?) cells were captured…was young was omitted in the legend?

There were not enough young Pax7+S100B+ cells to include in the violin plots in Figure 3 Sup 2A. As such, we made violin plots only from aged cells in the supplemental figure.